# Native American ataxia medicines rescue ataxia-linked mutant potassium channel activity via binding to the voltage sensing domain

Rían W. Manville[1], J. Alfredo Freites [2], Richard Sidlow[3], Douglas J. Tobias[2] & Geoffrey W. Abbott [1] ✉

There are currently no drugs known to rescue the function of Kv1.1 voltage-gated potassium channels carrying loss-of-function sequence variants underlying the inherited movement disorder, Episodic Ataxia 1 (EA1). The Kwakwa-ka'wakw First Nations of the Pacific Northwest Coast used *Fucus gardneri* (bladderwrack kelp), *Physocarpus capitatus* (Pacific ninebark) and *Urtica dioica* (common nettle) to treat locomotor ataxia. Here, we show that extracts of these plants enhance wild-type Kv1.1 current, especially at subthreshold potentials. Screening of their constituents revealed that gallic acid and tannic acid similarly augment wild-type Kv1.1 current, with submicromolar potency. Crucially, the extracts and their constituents also enhance activity of Kv1.1 channels containing EA1-linked sequence variants. Molecular dynamics simulations reveal that gallic acid augments Kv1.1 activity via a small-molecule binding site in the extracellular S1-S2 linker. Thus, traditional Native American ataxia treatments utilize a molecular mechanistic foundation that can inform small-molecule approaches to therapeutically correcting EA1 and potentially other Kv1.1-linked channelopathies.

The term 'ataxia' describes a category of neurological disorders comprising altered balance, coordination, gait and speech[1]. Episodic ataxia 1 (EA1) is an autosomal dominant inherited form caused by genetic variation in the human *KCNA1* gene, which encodes the Kv1.1 voltage-gated potassium (Kv) channel[2,3]. Affected individuals are typically heterozygous, bearing one wild-type and one mutant allele. EA1 alters central and peripheral nerve function, reflecting the role of Kv1.1 in both the central nervous system (especially cerebellum, hippocampus and neocortex) and peripheral nervous systems (especially synaptic terminal sites and juxtaparanodal regions of the nodes of Ranvier of myelinated axons). Most EA1-linked *KCNA1* mutations impair Kv1.1 function by directly disrupting its gating or ion conduction[4,5].

Kv1.1 channel activity tends to dampen neuronal excitability, and Kv1.1 loss-of-function mutations such as those in EA1, therefore, have the opposite effect, lowering the threshold for action potential generation. This can result in increased firing frequency, broadening of individual action potentials, and increased neurotransmitter release[6]. Kv channels form as tetramers of pore-forming α subunits, with heteromeric formation often occurring, typically with isoforms of the same subfamily. Kv1.1 can form homomeric channels in vitro, but is thought to almost exclusively form heteromeric channels in neurons in vivo, e.g., with Kv1.2; one exception is that homomeric Kv1.1 occurs in demyelinated neurons, such as in multiple sclerosis[7–11]. The normal function of Kv1.1/Kv1.2 channels at juxtaparanodal regions and branch points of myelinated axons helps maintain normal neuromuscular

[1]Bioelectricity Laboratory, Dept. of Physiology and Biophysics, School of Medicine, University of California, Irvine, CA, USA. [2]Department of Chemistry, University of California, Irvine, CA, USA. [3]Valley Children's Hospital, Madera, CA, USA. ✉e-mail: abbottg@hs.uci.edu

transmission and limit abnormal axonal firing[12–14], with these controls being compromised by pharmacological inhibition or by loss-of-function mutations[10].

Episodic ataxia is a relatively rare condition, affecting an estimated 26/100,000 people worldwide[15]; there are 9 reported inherited forms, with EA1 (*KCNA1*) and EA2 (which is linked to the calcium channel gene, *CACNA1A*) being the most common[16,17]. Episodic ataxia symptoms vary between forms and even within forms but generally involve recurring episodes of poor coordination and balance. The episodes can also involve blurred vision, slurred speech, vertigo, nausea and emesis, migraines, tinnitus, muscle weakness, hemiplegia and seizures[18]. Some individuals with episodic ataxia also exhibit myokymia (especially in the interictal interval in EA1), characterized by stiffness, muscle cramping, and fine muscle twitching that gives the appearance of subcutaneous rippling[2].

Present approaches to treating EA1 and EA2 typically involve administration of anticonvulsant/antiseizure medications, especially carbamazepine for EA1[19]. Acetazolamide is also used to treat EA1 but is more effective at treating EA2[15,20,21]. Retigabine, the first-in-class anticonvulsant that works by augmenting activity of the KCNQ2/3 (Kv7.2/3) Kv channel (which generates the neuronal M current), is an effective anticonvulsant but was withdrawn in 2017 because of side effects unrelated to KCNQ2/3 channel opening. It is, however, still being used off-label for pediatric patients with KCNQ2 epileptic encephalopathy, a severe disorder involving developmental delay and seizures[22]. Unlike KCNQ2/3, for which a plethora of direct-binding channel openers are known, Kv1.1 openers are rare. Pimaric acid, the first reported Kv1.1 channel opener, increases wild-type Kv1.1 currents up to eightfold (prepulse current at −50 mV) at 10 μM but higher concentrations inhibit Kv1.1. Pimaric acid also activates many other Kv channel isoforms, yet has not been shown to rescue EA1-linked Kv1.1 mutant channels[23]. We previously found that several synthetic derivatives of glycine augment wild-type Kv1.1 tail currents as much as 30-fold (after a −60 mV prepulse), with submicromolar potencies, no inhibition at higher concentrations, and no effects on Kv1.2 (and in some cases no effects on KCNQ2/3). However, even the most effective and selective glycine-derived Kv1.1 opener exhibited negligible rescue of the EA1 mutant Kv1.1 channels we tested (E283K, G311D, L328V, and V408A)[24].

We recently found multiple examples of KCNQ channel activation by extracts from plants, many of which help to explain the therapeutic effects for which the plants are traditionally used[25–31]. Given the absence of small molecules known to rescue EA1 loss-of-function mutant Kv1.1 channel activity, we turned to traditional botanical medicine. Here, we report rescue of Kv1.1 EA1 mutant channel activity by Native American folk medicines traditionally used to treat ataxia, and their constituent compounds. We also report discovery of the target site, a small-molecule-binding site in the S1-S2 linker of the Kv1.1 voltage sensing domain (VSD).

## Results

### Screening of Native American botanical medicines reveals wild-type Kv1.1 openers

Documented Native American folk medicine treatments of ataxia are, as expected, much less frequently represented in the literature than treatment for more common conditions, such as infections, pain, and gastrointestinal conditions. However, Chapman Turner and Bell's 1973 ethnobotanical study of the Southern Kwakwaka'wakw First Nation of what is now British Columbia included a description of their use of *Urtica dioica* (common nettle), *Physocarpus capitatus* (Pursh) Kuntze (Pacific ninebark) root extract and *Fucus gardneri* Silva (bladderwrack kelp) (Fig. 1a) as treatments for locomotor ataxia[32].

We therefore first tested for activity of extracts (diluted 1:50 in bath solution to equal 5 mg plant solid starting material per ml) of each of the three plants on wild-type human Kv1.1 and closely related Kv1.2

expressed in *Xenopus laevis* oocytes, using manual two-electrode voltage clamp (TEVC). Strikingly, all three extracts altered Kv1 channel activity. Nettle extract negative-shifted the voltage dependence and increased the maximal tail current of both Kv1.1 and Kv1.2. Pacific ninebark (roots or leaves) had similar effects on Kv1.1 but negligible effects on Kv1.2. Kelp extract was overall inhibitory upon both Kv1.1 and Kv1.2 peak currents at depolarized potentials, but negative-shifted the voltage dependence of activation of Kv1.1 by −17.4 ± 1.0 mV, and Kv1.2 by −18.1 ± 2.4 mV, and also steepened the slope of voltage dependence of activation -twofold for Kv1.1 (from 6.76 ± 0.67 to 3.30 ± 1.04 mV ($p = 0.027$; $n = 5$), but not Kv1.2 (Fig. 1b, c; Supplementary Information) (For all values and statistics accompanying the data in Figs. 1–10, see Supplementary Information). In addition, kelp, nettle and ninebark extracts in most cases hyperpolarized the resting membrane potential of *Xenopus* oocytes expressing Kv1.1 or Kv1.2 (Fig. 1d).

### Plant extract components gallic acid, tannic acid, and rutin each open wild-type Kv1.1

We next screened compounds known to be in kelp, nettle and/or ninebark[33–35], for their ability to open wild-type Kv1.1 potassium channels. Of seven compounds tested at 100 μM, gallic acid and tannic acid were each highly efficacious Kv1.1 channel openers, inducing constitutive activation at −80 mV, shifting the midpoint voltage-dependence of activation ($V_{0.5\ act}$) by −21.0 ± 2.5 mV and −25.6 ± 2.6 mV, respectively, and increasing prepulse currents at hyperpolarized potentials and tail current across the voltage range tested (Fig. 2a–c).

Nettle (*Urtica dioica*) leaves contain both gallic acid and hydrolysable tannins, the class of compounds that contains tannic acid and similar molecules that when hydrolyzed generate gallic or ellagic acids[36,37].

*Fucus* species contain gallic acid; with respect to tannins, like other brown seaweeds their tannin content is dominated by phlorotannins, with little to no condensed tannins, or hydrolysable tannins such as tannic acid[38,39]. Plants in the *Rosaceae* family, which includes Pacific ninebark (*Physocarpus capitatus*), contain both gallic acid and hydrolysable tannins[40]. Here, we confirmed the presence of gallic acid specifically in *Fucus gardneri* and *P. capitatus*, using Reversed Phase High-Performance Liquid Chromatography (RPHPLC) and mass spectroscopy (MS) (Supplementary Fig. 1).

Rutin, which is present in nettles[35] and ninebark[33] had a lesser but still robust opening effect, especially at hyperpolarized potentials and particularly apparent in the tail current. Cytisine, catechin hydrate, kaempferol, and quercetin did not alter Kv1.1 activity (Fig. 2a–c). Consistent with these data, gallic acid, tannic acid and rutin, but not the other compounds, each hyperpolarized the resting membrane potential of oocytes expressing Kv1.1 (Fig. 2d).

The impressive efficacy of gallic acid and tannic acid in opening Kv1.1 suggested that bark extracts, some of which contain one or both of these and also related compounds, and many of which are staples of Native American botanical medicine, might also be effective Kv1.1 openers. This hypothesis was confirmed by a further screen, in which we found that in particular white oak (*Quercus alba*), and to a lesser extent white willow (*Salix alba*), and cramp bark (*Viburnum opulus*), but not wild cherry bark (*Prunus serotina*), were able to augment Kv1.1 activity by negative-shifting its voltage dependence of activation (Fig. 3a, b) and also hyperpolarize the resting membrane potential ($E_M$) of oocytes expressing Kv1.1 (Fig. 3c). Importantly, of the barks tested here, we previously found that white oak is the only extract with appreciable gallic acid content[27]. Similarly, rutin and quercetin are also present in extracts from other folk medicine traditions, including *Sophora japonica*, a plant used in traditional Chinese, Japanese and Korean medicine with the intention of improving cardiovascular, brain and immune function; oxymatrine is also thought to be an active component of *S. japonica* extract[41]. We therefore tested *S. japonica*

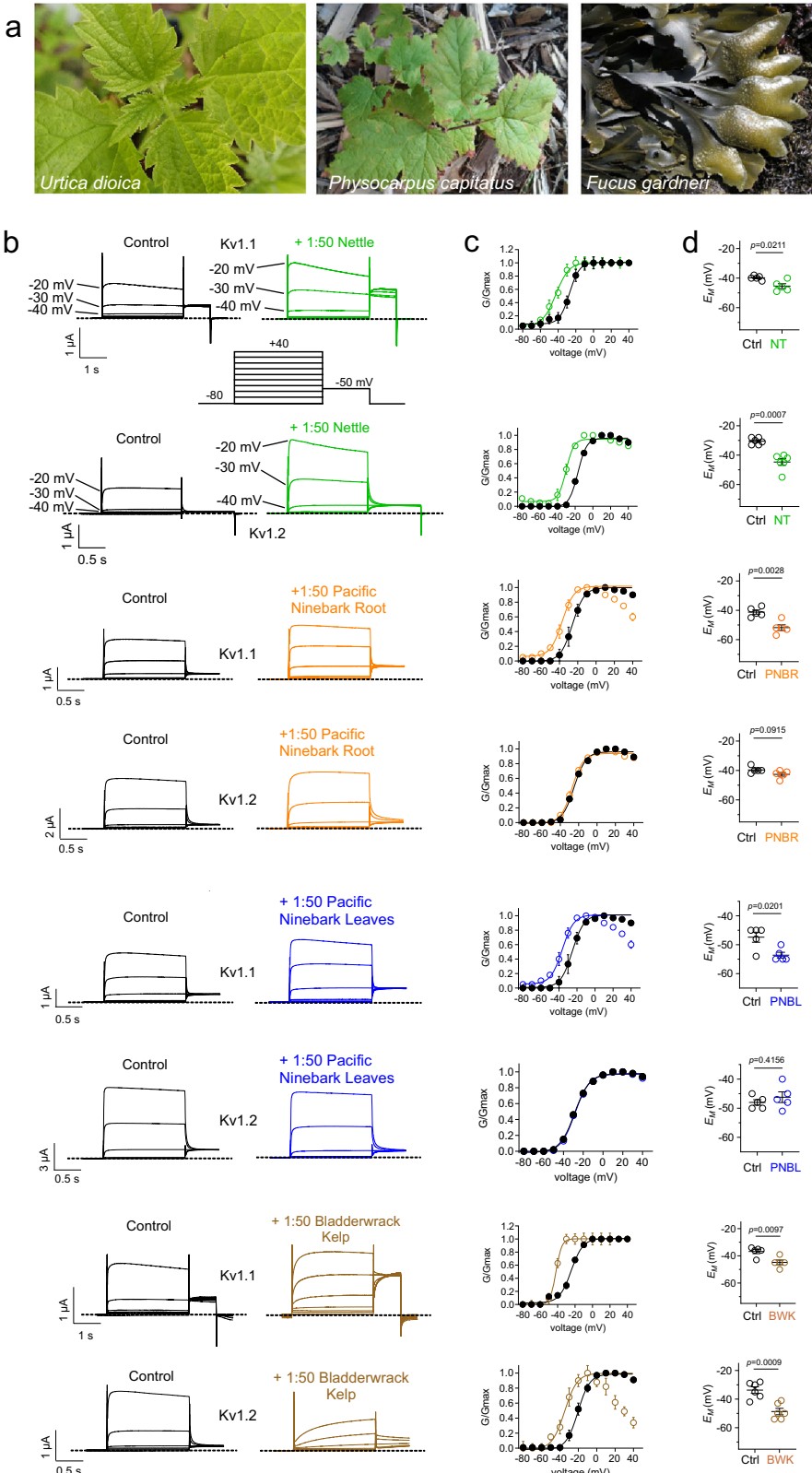

extract and oxymatrine and found that the extract activates Kv1.1 with an efficacy and characteristics similar to that which we observed for nettle extract (Fig. 1c–e), while oxymatrine had no effect, suggesting that rutin but not oxymatrine is an active component in this case (Fig. 3a, b). Similarly, *S. japonica* extract but not oxymatrine hyperpolarized the resting membrane potential of Kv1.1-expressing oocytes (Fig. 3c).

## Gallic acid and tannic acid are potent and efficacious wild-type Kv1.1 openers

Gallic acid markedly increased wild-type Kv1.1 prepulse and tail currents, especially at hyperpolarized potentials (Fig. 2a–c). Dose response studies revealed that gallic acid is a potent and efficacious Kv1.1 opener, with an $EC_{50}$ for $\Delta V_{0.5activation}$ of $54 \pm 10$ nM, and a maximal $-20$ mV shift in $V_{0.5activation}$. Strikingly, this effect was isoform-selective, with Kv1.2

**Fig. 1 | Plant extracts used by Kwakwaka'wakw for ataxia therapy enhance activation at negative potentials of wild-type Kv1.1 and/or Kv1.2.** Error bars indicate SEM. $n$ indicates number of biologically independent oocytes. At least 2 batches of oocytes were used per experiment. Statistical comparisons by two tailed paired $t$-test. Dashed lines indicate zero current line here and throughout. Source data are available. **a** Plants from which extracts were used by Kwakwaka'wakw for ataxia therapy. Photo credits: *Urtica dioica* (nettle)–Bo Abbott (used by permission). *Physocarpus capitatus* (Pacific ninebark)–senior author (GWA). *Fucus gardneri* (Bladderwrack kelp)–Steve Lonhart/NOAA MBNMS–http://sanctuarymonitoring. org/photos/photo_info.php?photoID=3530&search=algae&s=260&page=14, Public Domain, https://commons.wikimedia.org/w/index.php?curid=40550057. **b** Mean traces for Kv1.1 or Kv1.2 as indicated expressed in oocytes in the absence (Control)

or presence of 1:50 dilution extracts as indicated. Scale bars lower left for each pair of traces; voltage protocol upper inset; $n = 4$–6 per group. **c** Mean normalized tail current ($G/G_{max}$) for traces as in **b**. Kv1.1 nettles ($n = 5$); Kv1.2 nettles ($n = 6$); Kv1.1 1:50 pacific ninebark root ($n = 5$); Kv1.2 pacific ninebark root ($n = 5$); Kv1.1 pacific ninebark leaves ($n = 5$); Kv1.2 pacific ninebark leaves ($n = 5$); Kv1.1 bladderwrack kelp ($n = 4$); Kv1.2 bladderwrack kelp ($n = 6$). **d** Mean unclamped oocyte membrane potential for oocytes as in B; Kv1.1 nettles ($n = 5$ per; $p = 0.0211$); Kv1.2 nettles ($n = 6$; $p = 0.0007$); Kv1.1 pacific ninebark root ($n = 5$; $p = 0.0028$); Kv1.2 pacific ninebark root ($n = 5$; $p = 0.0915$); Kv1.1 pacific ninebark leaves ($n = 5$; $p = 0.0201$); Kv1.2 pacific ninebark leaves ($n = 5$; $p = 0.4156$); Kv1.1 bladderwrack kelp ($n = 4$; $p = 0.0097$); Kv1.2 bladderwrack kelp ($n = 6$; $p = 0.0009$).

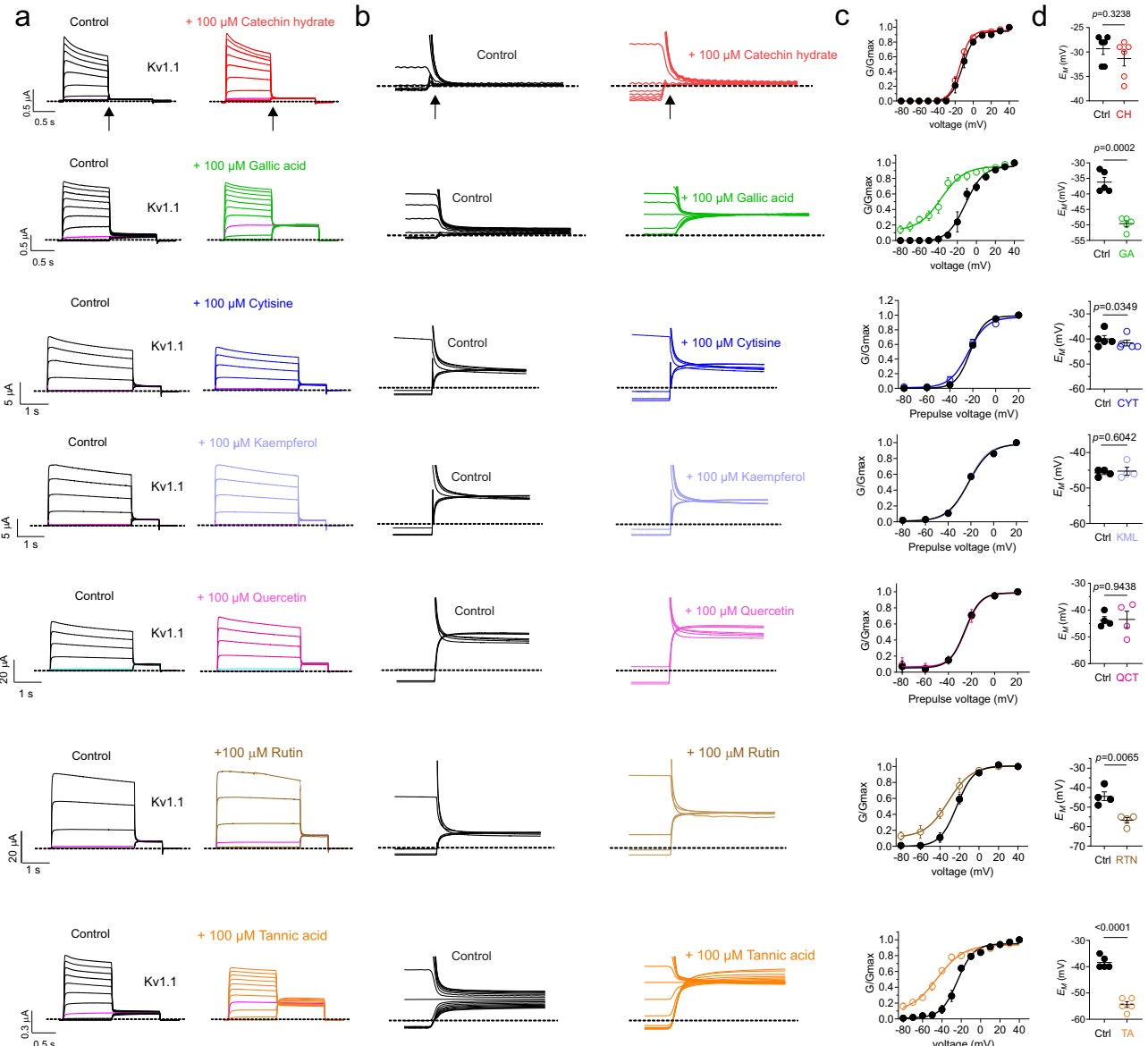

**Fig. 2 | Identification of constituents of plant extracts used by Kwakwaka'-wakw for ataxia therapy that enhance activation at negative potentials of wild-type Kv1.1.** Error bars indicate SEM. $n$ indicates number of biologically independent oocytes. At least 2 batches of oocytes were used per experiment. Statistical comparisons by two tailed paired t-test. Source data are available. **a** Mean traces for Kv1.1 expressed in oocytes in the absence (Control) or presence of plant extract components (100 μM) as indicated. Scale bars lower left for each trace; voltage protocol upper inset; $n = 4$–6 per group. Arrow indicates where tail current measurements are made for $G/G_{max}$ plots. Highlighted traces (magenta or cyan) show same-voltage pulses in each pair for comparison. **b** Close up of tail

currents from **a**; arrow indicates where tail current measurements are made for $G/G_{max}$ plots. **c** Mean normalized tail current ($G/G_{max}$) for traces as in **a**. 100 μM catechin hydrate ($n = 5$); 100 μM gallic acid ($n = 5$); 100 μM cytisine ($n = 5$); 100 μM kaempferol ($n = 4$); 100 μM quercetin ($n = 4$); 100 μM rutin ($n = 4$); 100 μM tannic acid ($n = 5$). **d** Mean unclamped oocyte membrane potential for oocytes as in **a**; 100 μM catechin hydrate ($n = 5$; $p = 0.3238$); 100 μM gallic acid ($n = 5$; $p = 0.0002$); 100 μM cytisine ($n = 5$; $p = 0.0349$); 100 μM kaempferol ($n = 4$; $p = 0.6042$); 100 μM quercetin ($n = 4$; $p = 0.9438$); 100 μM rutin ($n = 4$; $p = 0.065$); 100 μM tannic acid ($n = 5$; $<0.0001$).

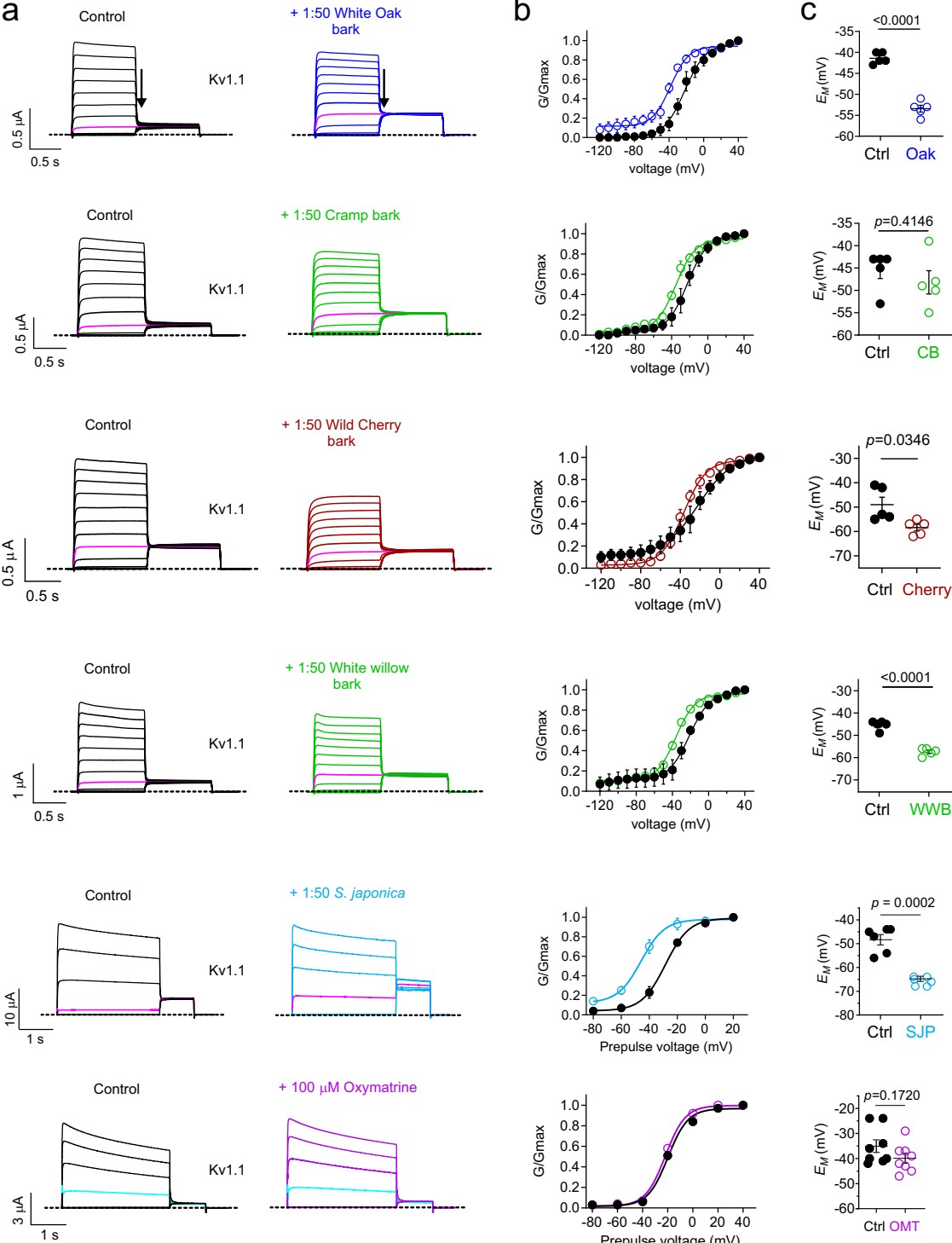

**Fig. 3 | Identification of other medicinal plant extracts with wild-type Kv1.1 function-enhancing activity.** Error bars indicate SEM. $n$ indicates number of biologically independent oocytes. At least 2 batches of oocytes were used per experiment. Statistical comparisons by two tailed paired $t$-test. Source data are available. **a** Mean traces for Kv1.1 expressed in oocytes in the absence (Control) or presence of plant extract (1:50) as indicated. Scale bars lower left for each trace; voltage protocol as in Fig. 2; $n = 4$–8 per group. Arrow indicates where tail current measurements are made for $G/G_{max}$ plots. Highlighted traces (magenta or cyan)

show same-voltage pulses in each pair for comparison. **b** Mean normalized tail current ($G/G_{max}$) for traces as in **a**. White oak bark ($n = 5$); cramp bark ($n = 5$); wild cherry bark ($n = 5$); white willow bark ($n = 5$); *Saphora japonica* ($n = 6$); 100 μM oxymatrine ($n = 8$). **c** Mean unclamped oocyte membrane potential for oocytes as in **a**; white oak bark ($n = 5$; <0.0001); cramp bark ($n = 5$; $p = 0.4146$); wild cherry bark ($n = 5$; $p = 0.0346$); white willow bark ($n = 5$; <0.0001); *Saphora japonica* ($n = 6$; $p = 0.0002$); 100 μM oxymatrine ($n = 8$; $p = 0.1720$).

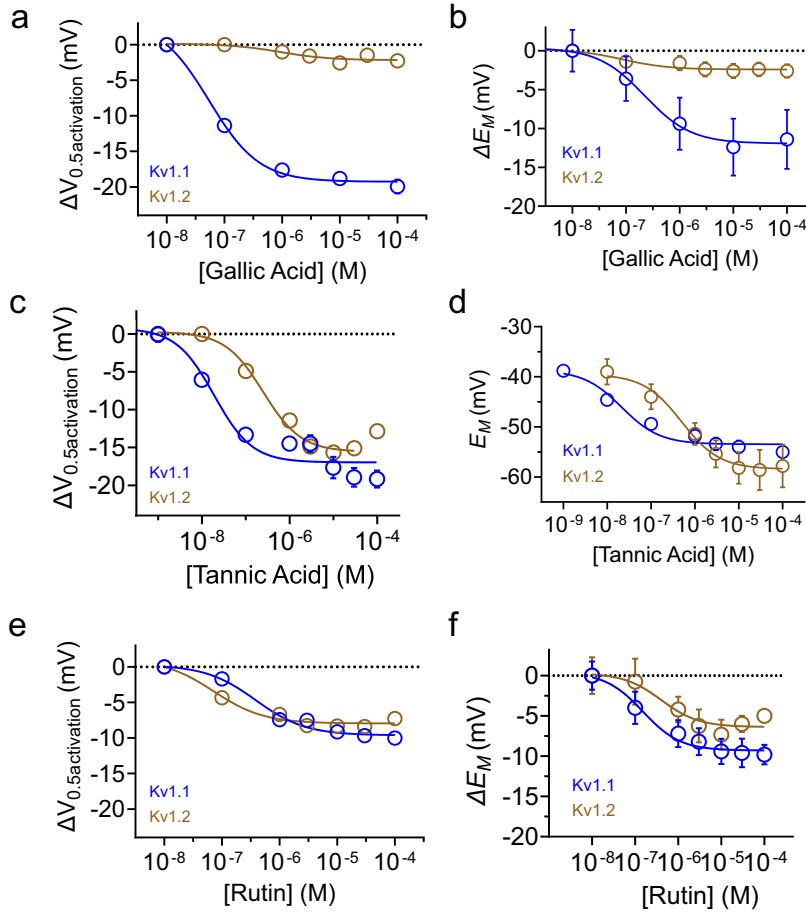

**Fig. 4 | Dose responses for wild-type Kv1.1 and Kv1.2 modulation by gallic acid, tannic acid and rutin.** Error bars indicate SEM. $n$ indicates number of biologically independent oocytes. At least 2 batches of oocytes were used per experiment. Source data are available. **a** Mean shift in voltage dependence of activation ($\Delta V_{0.5\text{activation}}$) vs. [gallic acid] for Kv1.1 ($n = 5$) and Kv1.2 ($n = 5$), quantified from traces similar to those shown in Fig. 2. **b** Mean change in unclamped oocyte membrane potential ($\Delta E_{\text{M}}$) vs. [gallic acid] for oocytes expressing Kv1.1 ($n = 5$) and Kv1.2 ($n = 5$). **c** Mean $\Delta V_{0.5\text{activation}}$ vs. [tannic acid] for Kv1.1 and Kv1.2, quantified from traces similar to those shown in Fig. 2. Kv1.1 tannic acid concentrations being relatively insensitive (Fig. 4a). Concentrations as low as 100 nM gallic acid begin to hyperpolarize the resting membrane potential of Kv1.1-expressing oocytes, with the $EC_{50}$ for cell membrane hyperpolarization being $0.2 \pm 0.6\,\mu M$, again with minimal effects on Kv1.2-expressing oocytes (Fig. 4b). Tannic acid was a similarly efficacious and potent ($EC_{50}$ for $\Delta V_{0.5\text{activation}} = 19 \pm 1.5$ nM) Kv1.1 opener but was less Kv1.1-selective, opening Kv1.2 with comparable efficacy, albeit with reduced potency ($EC_{50}$ for $\Delta V_{0.5\text{activation}} = 253 \pm 10$ nM) (Fig. 4c). Tannic acid potently and efficaciously hyperpolarized the $E_{\text{M}}$ of oocytes expressing Kv1.1 or Kv1.2 (Fig. 4d). Rutin did not differentiate well between Kv1.1 and Kv1.2 and was less efficacious in opening either channel, or hyperpolarizing oocytes expressing either channel, compared to gallic or tannic acid. However, rutin displayed relatively high potency ($EC_{50}$ of $385 \pm 12$ nM vs. $72 \pm 16$ nM, respectively, for Kv1.1 vs. Kv1.2 $\Delta V_{0.5\text{activation}}$; $EC_{50}$ of $150 \pm 38$ nM vs. $406 \pm 62$ nM, respectively, for hyperpolarization of $E_{\text{M}}$ of oocytes expressing Kv1.1 vs Kv1.2) (Fig. 4e, f).

0.01 μM ($n = 5$); 0.1 μM ($n = 12$); 1 μM ($n = 12$); 3 μM ($n = 5$); 10 μM ($n = 7$); 50 μM ($n = 7$); 100 μM ($n = 7$); 250 μM ($n = 6$); 500 μM ($n = 7$); Kv1.2 ($n = 5$). **d** Mean change in unclamped oocyte membrane potential ($\Delta E_{\text{M}}$) vs. [tannic acid] for oocytes expressing Kv1.1 or Kv1.2. Kv1.1 tannic acid concentrations 0.01 μM ($n = 5$); 0.1 μM ($n = 12$); 1 μM ($n = 12$); 3 μM ($n = 5$); 10 μM ($n = 7$); 50 μM ($n = 7$); 100 μM ($n = 7$); 250 μM ($n = 6$); 500 μM ($n = 7$); Kv1.2 ($n = 5$). **e** Mean $\Delta V_{0.5\text{activation}}$ vs. [rutin] for Kv1.1 ($n = 5$) and Kv1.2 ($n = 5$), quantified from traces similar to those shown in Fig. 2. **f** Mean change in unclamped oocyte membrane potential ($\Delta E_{\text{M}}$) vs. [rutin] for oocytes expressing Kv1.1 ($n = 5$) and Kv1.2 ($n = 5$).

## Native American ataxia remedies improve function of some inherited EA1 mutant Kv1.1 channels

We next evaluated the ability of Native American ataxia remedies to enhance or rescue the function of EA1-linked sequence variants of Kv1.1 channels expressed in *Xenopus* oocytes. We screened five gene variants: L155P, E283K, G311D, L328V and V408A, spanning the Kv1.1 N-terminal through to the C-terminal end of S6 (Fig. 5a). Two of the variants, L155P and E283K, showed responsiveness to one or more of the traditional ataxia remedy plant extracts (Figs. 5–6; Supplementary Fig. 2), while G311D, L328V and V408A were minimally responsive or nonresponsive (Supplementary Figs. 3–8).

L155P is the first reported EA1-linked variant located in the Kv1.1 N-terminal cytoplasmic domain (Fig. 5a). We recently demonstrated that L155P eradicates activity in 100% mutant channels and non-dominantly impairs activity in wild-type Kv1.1/Kv1.1-L155P channels generated from co-injection in oocytes of 50/50 wild-type/L155P Kv1.1 cRNA (hereafter referred to as "Kv1.1/Kv1.1-L155P" channels[42]. When applied to 100% Kv1.1-L155P channels (hereafter referred to as "Kv1.1-L155P" channels, the extracts were unable to rescue activity (Supplementary Fig. 9). In contrast, 1/50 bladderwrack kelp and Pacific ninebark leaf extracts shifted (by $-15.1 \pm 2.1$ and $-10.1 \pm 0.9$ mV; $P = 0.0005$ and $< 0.0001$, respectively, $n = 5$) Kv1.1/Kv1.1-L155P channel voltage dependence (Fig. 5b, c), and the $E_{\text{M}}$ (by $-11.4 \pm 0.8$ and $-9.6 \pm 0.2$ mV; $P = < 0.0001$ and $0.0036$, respectively, $n = 5$) of oocytes expressing Kv1.1/Kv1.1-L155P (Fig. 5d) (and see Supplementary Information).

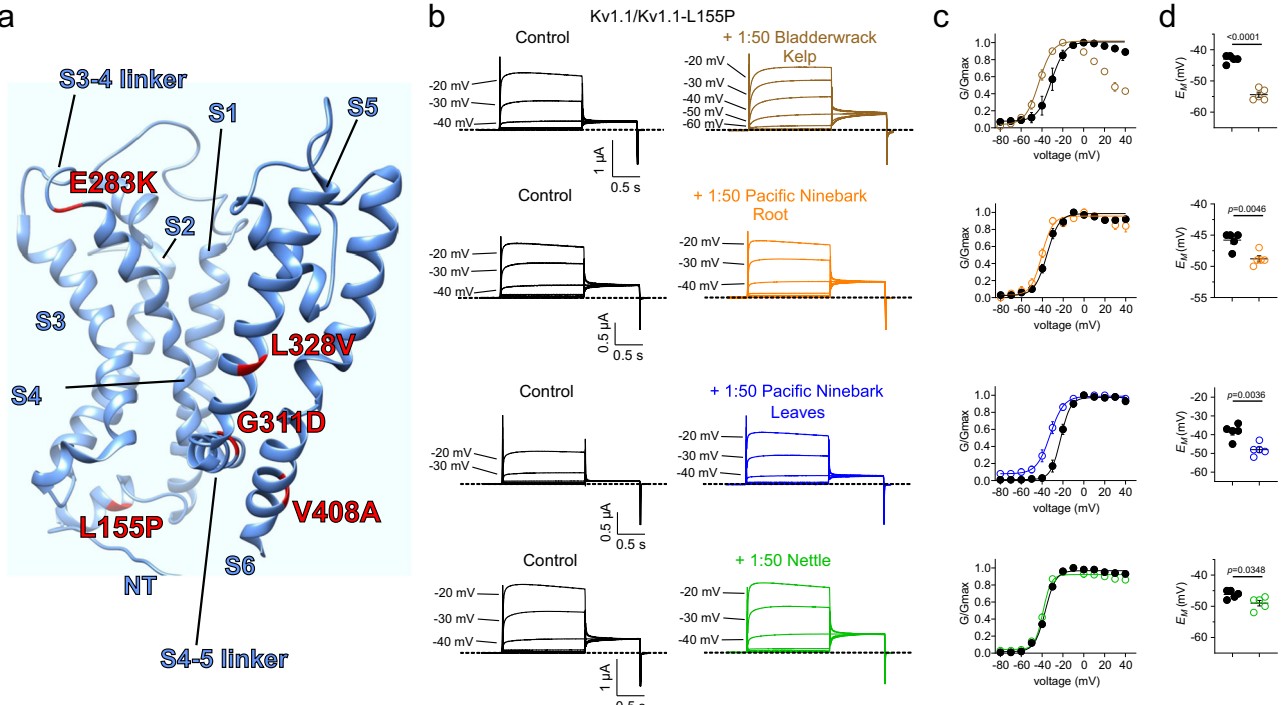

**Fig. 5 | Specific plant extracts used by Kwakwaka'wakw for ataxia therapy enhance activation of EA1-linked Kv1.1/Kv1.1-L155P channels.** Error bars indicate SEM. *n* indicates number of biologically independent oocytes. At least 2 batches of oocytes were used per experiment. Statistical comparisons by two-tailed paired *t*-test. Source data are available. **a** Schematic of location of EA1 sequence variants on Kv1.1 studied herein. **b** Mean traces for Kv1.1/Kv1.1-L155P channels expressed by co-injection of 50/50 wild-type/L155P Kv1.1 cRNA in oocytes in the absence (Control) or presence of 1:50 dilution extracts as indicated. Scale bars lower left for each trace; voltage protocol as in Fig. 2; *n* = 5 per group. **c** Mean normalized tail current ($G/G_{max}$) for traces as in **b**; *n* = 5 per group. **d** Mean unclamped oocyte membrane potential for oocytes as in **b**; bladderwrack kelp (*n* = 5; <0.0001); pacific ninebark root (*n* = 5; *p* = 0.0046); pacific ninebark leaves (*n* = 5; *p* = 0.0036); nettle (*n* = 5; *p* = 0.0348).

Kv1.1-E283K, an EA1 mutation situated in the S3-4 linker (Fig. 5a), shifts Kv1.1 $V_{0.5activation}$ by +10-20 mV and slows its activation twofold[11]. Strikingly, Pacific ninebark (roots or leaves), kelp and nettles were each able to restore Kv1.1-E283K channel voltage dependence to that of wild-type Kv1.1, shifting its voltage dependence back close to or even overlaying that of wild-type Kv1.1, as evident from $G/G_{max}$ plots with wild-type control $G/G_{max}$ plots (from Fig. 1) overlaid (Fig. 6a, b); the extract-induced shifts in Kv1.1-E283K $V_{0.5activation}$ ranged from −14 to −18 mV (Supplementary Information). Likewise, each of the extracts was influential on cellular excitability of oocytes expressing Kv1.1-E328K, shifting $E_M$ by between −10 and −17 mV (Fig. 6c; Supplementary Information). Bladderwrack kelp had additional effects on gating kinetics and so we examined this extract further. Kelp increased Kv1.1-E283K prepulse current >fivefold at −30 mV (Fig. 6d). While kelp extract further slowed Kv1.1-E283K activation (Fig. 6e), it also greatly slowed deactivation (Fig. 6f), which contributed to the substantially increased magnitude and longevity of the tail current recorded at −50 mV (Fig. 6a). The extracts were similarly effective at augmenting activity of Kv1.1/Kv1.1-E283K channels (Supplementary Fig. 10A–D).

EA1 mutant G311D is in the S4-5 linker of Kv1.1 (Fig. 5a) and produces a 65% prepulse current reduction both in Kv1.1-G311D and in Kv1.1/Kv1.1-G311D channels[43,44]; either channel type was minimally responsive to the plant extracts tested (Supplementary Figs. 3 and 4). EA1 mutant L328V (which is associated with tetany and hypomagnesemia) is located in the intracellular end of Kv1.1 S5 (Fig. 5a) and results in complete current loss in a dominant negative manner, i.e., Kv1.1-L328V and Kv1.1/Kv1.1-L328V channels are each nonfunctional[45]. The traditional ataxia remedy plant extracts were unable to rescue Kv1.1-L328V or Kv1.1/Kv1.1-L328V function (Supplementary Figs. 5 and 6). EA1 mutant V408A is in the intracellular-proximal end of Kv1.1 S6 (Fig. 5a) and reduces current, speeds deactivation, and increases C-type

inactivation of Kv1.1[46]. Here, Native American ataxia remedy extracts had relatively minor effects on Kv1.1-V408A activity, with nettle and Pacific ninebark (roots or leaves) inducing hyperpolarizing shifts in $E_M$, and Pacific ninebark leaves inducing a negative shift in the voltage dependence of activation, although this was not quantifiable due to the very small tail currents (Supplementary Fig. 7). Effects of the extracts on Kv1.1/Kv1.1-V408A activity were also relatively minor (Supplementary Fig. 8).

## Gallic acid and tannic acid improve function of some inherited EA1 mutant Kv1.1 channels

The data above establish gallic acid and tannic acid as potent and efficacious agonists of wild-type Kv1.1 (Figs. 2 and 4), and demonstrate that Native American ataxia folk remedies (especially Pacific ninebark and bladderwrack kelp) can rescue EA1-linked Kv1.1 mutant function (Figs. 5 and 6). We recently discovered synthetic compounds with similar potency and efficacy for augmenting wild-type Kv1.1 activity, that were unable to rescue EA1 mutant Kv1.1 channel activity (in conditions mimicking either the homozygous or the heterozygous condition), even at 100 μM, neither could they improve the ability of EA1 mutant Kv1.1 channels to hyperpolarize the resting cell membrane potential[24].

Here, gallic acid (1 μM) was ineffective at rescuing activity of 100% mutant L155P, G311D, L328V, or V408A Kv1.1 channels (Supplementary Figs. 9 and 11), and imparted only very minor functional improvements on Kv1.1-E283K (Supplementary Fig. 11A–E). We next tested effects on channels generated by co-injection of 50/50 wild-type/mutant Kv1.1 cRNA. Strikingly, gallic acid (1 μM) was able to hyperpolarize (by −15.5 ± 0.5 mV; *n* = 4; *P* < 0.0001) the $V_{0.5 act}$ of Kv1.1/Kv1.1-E283K channels, restoring voltage dependence typical of wild-type Kv1.1 (Fig. 7a–c), also inducing a −14 ± 0.7 mV hyperpolarization (*n* = 4;

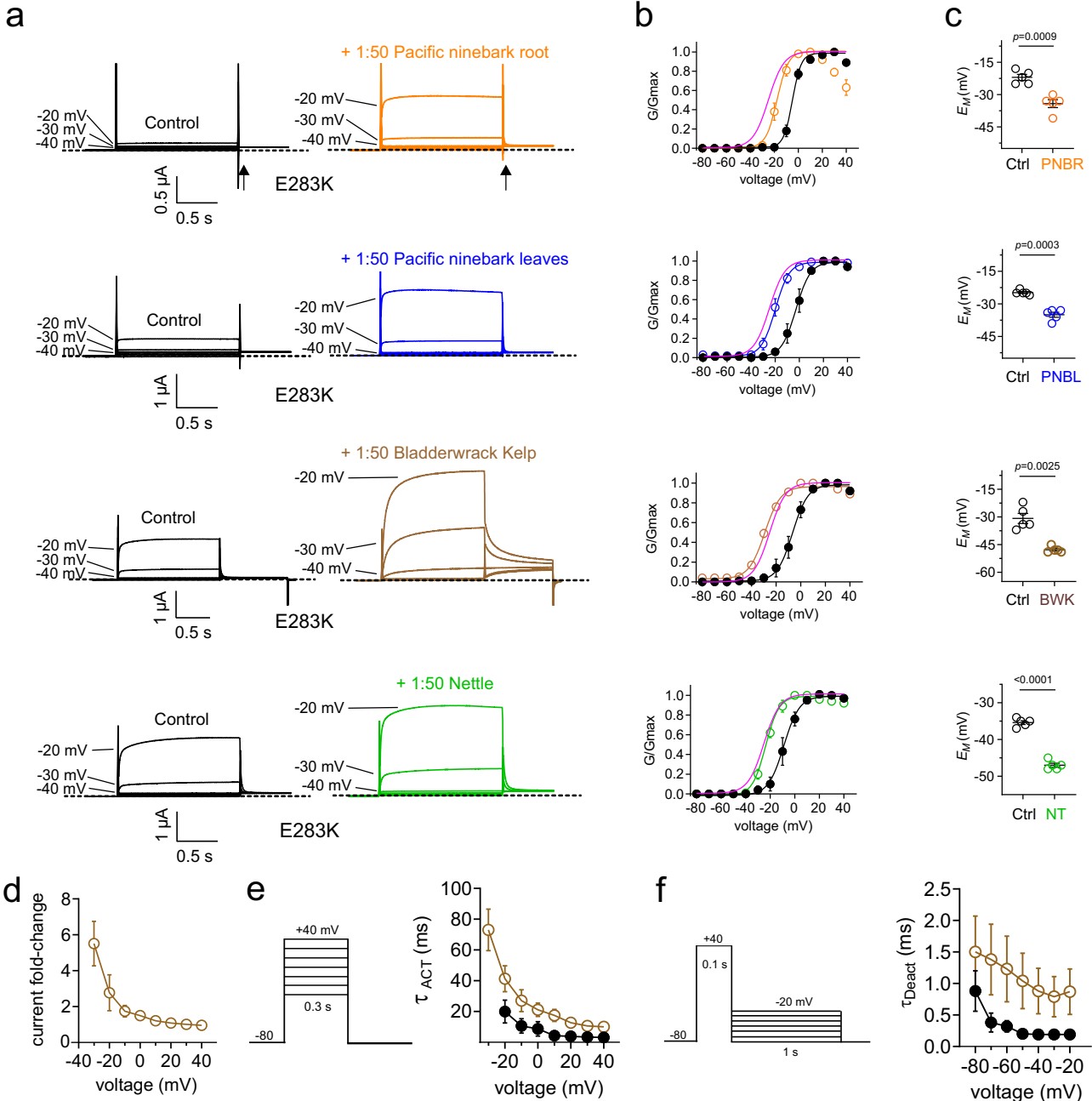

**Fig. 6 | Specific plant extracts used by Kwakwaka'wakw for ataxia therapy enhance activation of EA1-linked Kv1.1-E283K channels.** Error bars indicate SEM. *n* indicates number of biologically independent oocytes. At least 2 batches of oocytes were used per experiment. Statistical comparisons by two-tailed paired t-test. Source data are available. **a** Mean traces for Kv1.1-E283K channels expressed by injection of 100% Kv1.1-E283K cRNA in oocytes in the absence (Control) or presence of 1:50 dilution extracts as indicated (only −80 to −20 mV range shown, for clarity). Scale bars lower left for each trace; voltage protocol as in Fig. 2; *n* = 5 per group. Arrow indicates where tail current measurements are made for *G/G*max plots. **b** Mean normalized tail current (*G/G*max) for traces as in **a**; *n* = 5 per group.

Control wild-type Kv1.1 G/V curves (magenta) from Fig. 1 overlaid for comparison. **c** Mean unclamped oocyte membrane potential for oocytes as in **a**; pacific ninebark root (*n* = 5; *p* = 0.0009); pacific ninebark leaves (*n* = 5; *p* = 0.0003); bladderwrack kelp (*n* = 5; *p* = 0.0025); nettle (*n* = 5; <0.0001). **d** Mean Kv1.1-E283K peak prepulse current fold-change vs. voltage induced by bladderwrack kelp (1:50); *n* = 5. **e** Mean activation rate (*T*ACT) vs. voltage for Kv1.1-E283K in bath solution (black) vs. 1:50 bladderwrack kelp (brown). *Left*, voltage protocol; *n* = 5. **f** Mean deactivation rate (*T*Deact) vs. voltage for Kv1.1-E283K in bath solution (black) vs. 1:50 bladderwrack kelp (brown). *Left*, voltage protocol; *n* = 8.

$P = 0.0005$) in $E_M$ of oocytes expressing Kv1.1/Kv1.1-E283K (Fig. 7d). Effects on other 50/50 wild-type/mutant channels were negligible with gallic acid (1 μM) (Fig. 7a–d) except for a −6.8 ± 0.6 mV shift in the voltage dependence of activation of Kv1.1/Kv1.1-L155P (Fig. 8a–c) that resulted in a −9.1 ± 0.8 mV hyperpolarization of $E_M$ (Fig. 8d). Because of this hint at activity improvement in L155P channels we also applied higher concentrations (10 and 100 μM) of gallic acid, which produced a more robust (−12.5 ± 0.6 mV and −14.1 ± 0.6 mV, respectively)

hyperpolarization of Kv1.1/Kv1.1-L155P $V_{0.5\ act}$ (Fig. 8b, c) and hyperpolarized $E_M$ by −14.0 ± 0.6 mV and −15.2 ± 0.7 mV, respectively (Fig. 8d).

Kv1.1 α subunits form heteromeric tetramers with other Kv1 subfamily α subunits, including Kv1.2, in the extraparanodal region of axons and some terminals, in the hippocampus (perforant input from entorhinal cortex, mossy fiber projections from granule cells to CA3, and CA3 axon collaterals)[7–11]. We therefore tested effects

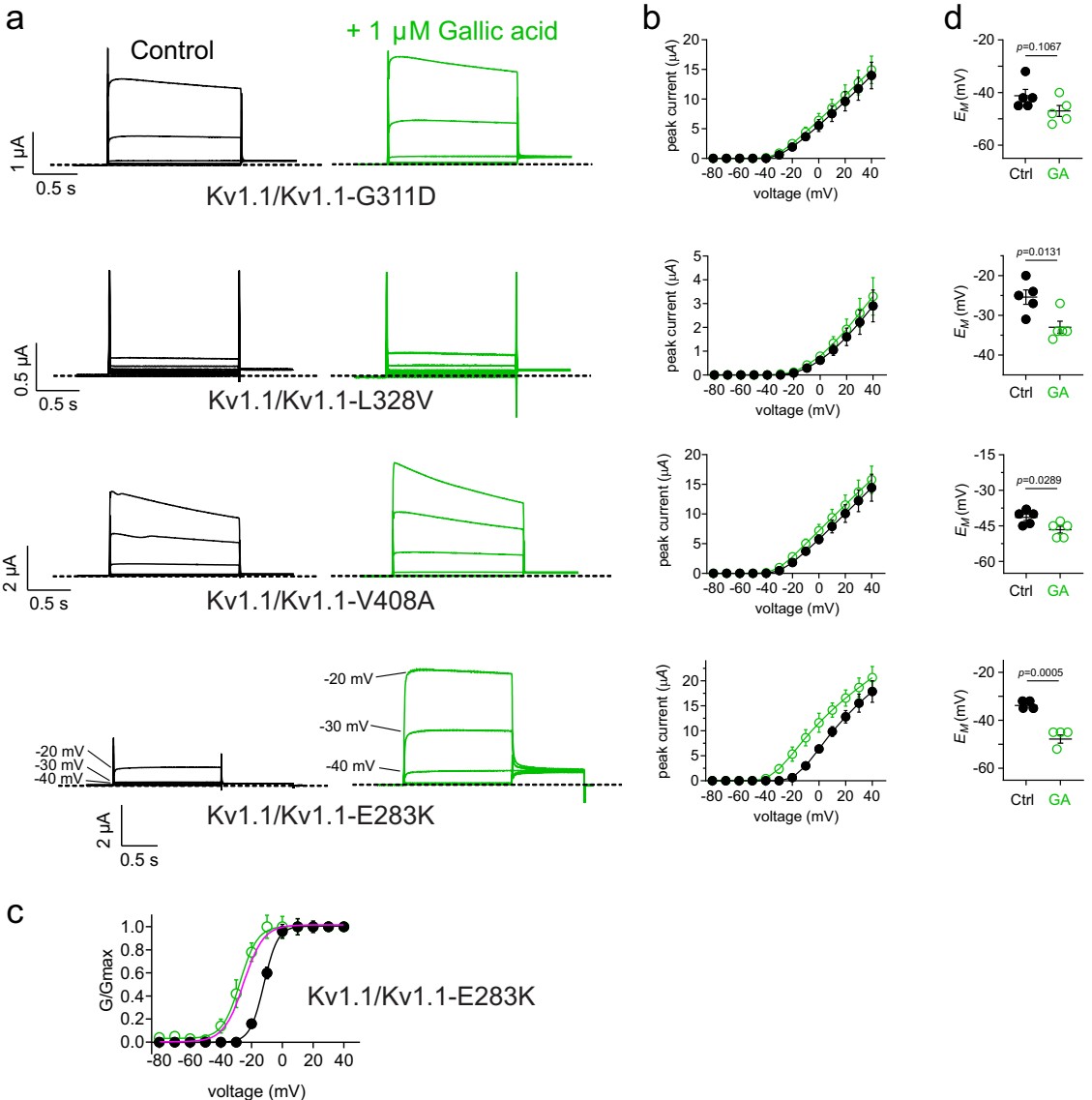

**Fig. 7 | Gallic acid enhances activation of EA1-linked Kv1.1/Kv1.1-E283K channels.** Error bars indicate SEM. *n* indicates number of biologically independent oocytes. At least 2 batches of oocytes were used per experiment. Statistical comparisons by two-tailed paired *t*-test. Source data are available. **a** Mean traces (only −80 to −20 mV range shown, for clarity) for EA1-linked mutant Kv1.1 channels as indicated, expressed by co-injection of 50/50 wild-type/mutant Kv1.1 cRNA in oocytes in the absence (Control) or presence of gallic acid (1 μM). Scale bars lower left for each trace; voltage protocol as in Fig. 2; *n* = 5 per group. **b** Mean peak current (measured during prepulse) from traces as in **a**; *n* = 5 per group. **c** Mean normalized tail current ($G/G_{max}$) for Kv1.1/Kv1.1-E283K traces as in **a**; *n* = 5 per group. Control wild-type Kv1.1 GV curve (magenta) from Fig. 1 overlaid for comparison. **d** Mean unclamped oocyte membrane potential for oocytes as in **a**; Kv1.1/Kv1.1-G311D (*n* = 5; *p* = 0.1067); Kv1.1/Kv1.1-L328V (*n* = 5; *p* = 0.0131); Kv1.1/Kv1.1-V408A (*n* = 5; *p* = 0.0289); Kv1.1/Kv1.1-E283K (*n* = 4; *p* = 0.0005).

of gallic acid on Kv1.1/Kv1.2 heteromers generated in oocytes by co-injection of 1:1:2 parts Kv1.1:Kv1.1-E283K:Kv1.2 or Kv1.1:Kv1.1-L155P:Kv1.2 cRNA to approximate the Kv1.1/Kv1.2 heteromeric channel composition in EA1 patients heterozygous for either mutant *KCNA1* allele. Expression of 50/50 wild-type/E283K Kv1.1 with wild-type Kv1.2 (Supplementary Fig. 12A) produced a positive shift in the voltage dependence of activation (Supplementary Fig. 12B) that was difficult to distinguish from the prepulse I/V relationship (Supplementary Fig. 12C) but more obvious from the raw and normalized tail currents (Supplementary Fig. 12D, E) and its depolarizing effect on $E_M$ (Supplementary Fig. 12F). Gallic acid was highly effective at rescuing Kv1.1/Kv1.1-E283K/Kv1.2 channel activity, even at 0.1 μM (Supplementary Fig. 12G), negative-shifting the voltage dependence of activation, hyperpolarizing $E_M$ (Supplementary Fig. 12H–K) and returning voltage dependence to that of wildtype Kv1.1/Kv1.2 heteromers (Supplementary Fig. 12L).

Expression of 50/50 wild-type/L155P Kv1.1 with wild-type Kv1.2 (1:1:2 cRNA ratio) (Supplementary Fig. 13A) reduced peak prepulse and tail current magnitude by ~50% (Supplementary Fig. 13A–D) with minimal change in voltage dependence of activation (Supplementary Fig. 13E) but a depolarizing effect on $E_M$ (Supplementary Fig. 13F). Gallic acid was again highly effective at rescuing Kv1.1/Kv1.1-L155P/Kv1.2 channel activity at 0.1 μM and higher (Supplementary Fig. 13G), increasing peak and tail current magnitude especially at negative potentials (Supplementary Fig. 13H, I) by negative-shifting the voltage dependence of activation (Supplementary Fig. 13J) and hyperpolarizing $E_M$ (Supplementary Fig. 13K).

Tannic acid (1 μM) was effective at increasing activity of Kv1.1-E283K especially at −40 to −20 mV (as much as 20-fold), negative-shifting Kv1.1-E283K $V_{0.5\ activation}$ by −13 ± 0.26 mV, and thus restoring a voltage dependence similar to that of wild-type Kv1.1 (Fig. 9a, b). This resulted in an almost −20 mV shift in the $E_M$ of oocytes expressing

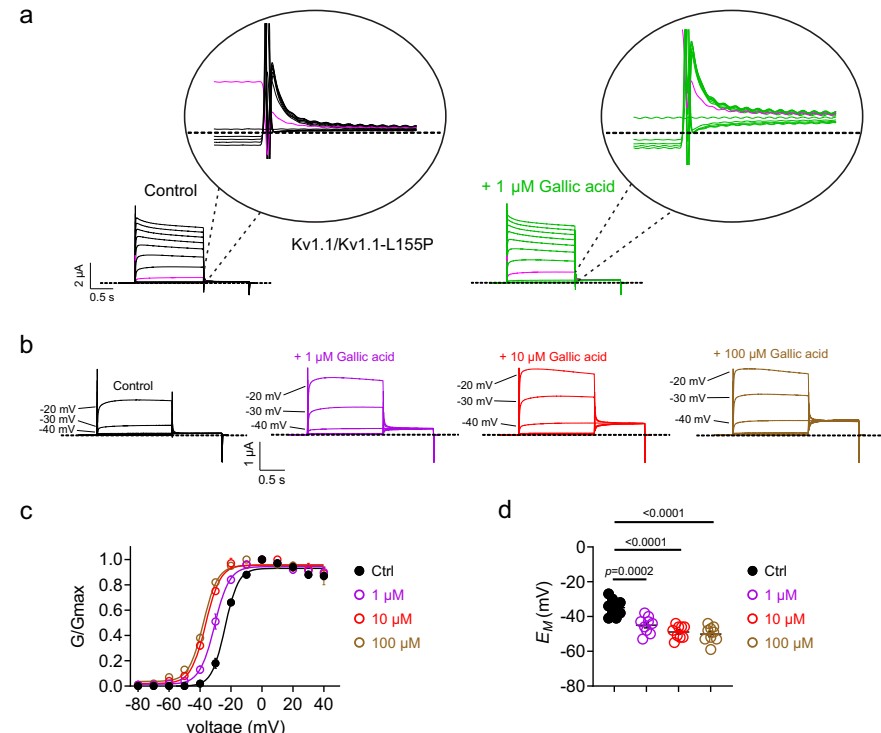

**Fig. 8 | Gallic acid enhances activation of EA1-linked Kv1.1/Kv1.1L155P channels.** Error bars indicate SEM. *n* indicates number of biologically independent oocytes. At least 2 batches of oocytes were used per experiment. Statistical comparisons by one-way ANOVA with post-hoc Tukey HSD. Source data are available. **a** Mean traces for Kv1.1/Kv1.1-L155P channels expressed by co-injection of 50/50 wild-type/L155P Kv1.1 cRNA in oocytes in the absence (Control) or presence of gallic acid (1 μM). Scale bars lower left; voltage protocol as in Fig. 2; *n* = 9 per group. Upper insets show close-up of tail currents. Magenta traces show same-voltage pulses in each

current family for comparison. **b** Mean traces for Kv1.1/Kv1.1-L155P channels expressed in oocytes in the absence (Control) or presence of gallic acid (1–100 μM). Scale bars lower left; voltage protocol as in Fig. 2; *n* = 9 per group. **c** Mean normalized tail current ($G/G_{max}$) from traces as in **a** and **b**; *n* = 9 per group. **d** Mean unclamped oocyte membrane potential for oocytes as in **a** and **b**; 1 μM gallic acid (*n* = 9; *p* = 0.0002); 10 μM gallic acid (*n* = 9; *p* < 0.0001); 100 μM gallic acid (*n* = 9; *p* < 0.0001).

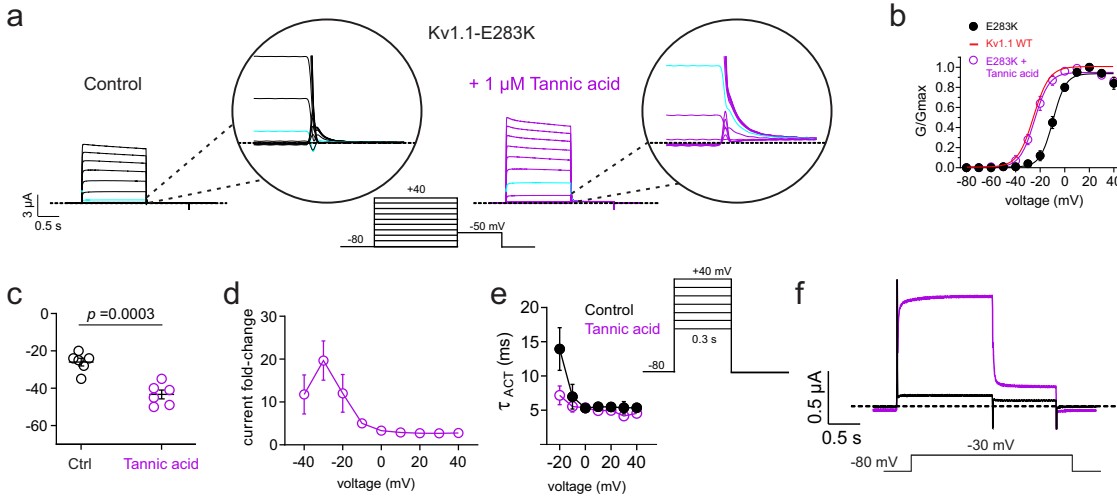

**Fig. 9 | Tannic acid enhances activation of EA1-linked Kv1.1-E283K channels.** Error bars indicate SEM. *n* indicates number of biologically independent oocytes. At least 2 batches of oocytes were used per experiment. Statistical comparisons by two-tailed paired t-test. Source data are available. **a** Mean traces for Kv1.1-E283K channels expressed by co-injection of 100% Kv1.1-E283K cRNA in oocytes in the absence (Control) or presence of tannic acid (1 μM). Scale bars lower left; voltage protocol lower inset; *n* = 6 per group. Upper insets show close-up of tail currents. *Cyan*, traces at −20 mV highlighted for ease of comparison. Cyan traces show same-voltage pulses in each current family for comparison. **b** Mean normalized tail

current ($G/Gmax$) from traces as in **a**; *n* = 6 per group. Control wild-type Kv1.1 $G/V$ curve (red) from Fig. 1 overlaid for comparison. **c** Mean unclamped oocyte membrane potential for oocytes as in **a** (*n* = 6; *p* = 0.0003). **d** Mean Kv1.1-E283K current fold-change vs. voltage induced by tannic acid (1 μM); *n* = 5. **e** Mean activation rate ($T_{ACT}$) vs. voltage for Kv1.1-E283K in bath solution (black) vs. tannic acid (1 μM) (purple). Voltage protocol, upper inset; *n* = 5. **f** Mean traces using voltage protocol indicated (upper inset) for Kv1.1-E283K in bath solution (black) vs. tannic acid (1 μM) (purple) to illustrate the dramatic current enhancing effects at −30 mV; *n* = 6.

Kv1.1-E283K (Fig. 9c). The $EC_{50}$ for effects of tannic acid on Kv1.1-E283K was calculated from the increase in relative tail current at −40 mV (Fig. 9d), resulting in a biphasic dose response; the lower sensitivity component $EC_{50}$ was not quantifiable because it did not saturate even at high [tannic acid] but the higher sensitivity component had an $EC_{50}$ of $3.7 ± 2.6$ nM (Fig. 9d; Supplementary Fig. 14A–C). Similarly, the effects of tannic acid on $E_M$ of oocytes expressing Kv1.1-E283K also showed a biphasic dose response, with $EC_{50}$ values of $1.1 ± 0.9$ nM and $10 ± 0.4$ μM mV, respectively (Supplementary Fig. 14D). Tannic acid slightly speeded Kv1.1-E283K activation at negative potentials (Fig. 9e); effects on peak and tail current magnitude are particularly evident in the mean traces at −30 mV (Fig. 9f).

Tannic acid also hyperpolarized the voltage dependence of activation (by −12 mV) of Kv1.1/Kv1.1-E283K (Supplementary Fig. 15A–C) and the $E_M$ of oocytes expressing Kv1.1/Kv1.1-E283K (Supplementary Fig. 13D); effects of tannic acid on Kv1.1 channels generated by expressing 50/50 wild-type/mutant (V408A, G311D, L155P or L328V) Kv1.1 were minimal (Supplementary Fig. 15A–D). Tannic acid (1 μM) imparted subtle improvements in Kv1.1-V408A function but slowed its activation (Supplementary Fig. 16). Tannic acid (1 μM) did not alter Kv1.1-L155P or Kv1.1-G311D activity; it increased peak current at 0 mV and positive membrane potentials in oocytes expressing Kv1.1-G311D but given that Kv1.1-G311D currents are <200 nA at 40 mV, this effect is highly likely to involve weak activation of endogenous oocyte currents, and this was insufficient to hyperpolarize $E_M$ (Supplementary Fig. 17). Rutin (1 μM) was ineffective at rescuing Kv1.1 EA1 mutant activity (Supplementary Fig. 18A–D).

### Gallic acid augments Kv1.1 subthreshold currents via a VSD-binding site

The gallic acid augmentation of both wild-type and EA1-linked mutant Kv1.1 channel activity is of particular interest because gallic acid is present in the plants Native Americans used for ataxia therapy, it is a small molecule sold over the counter and used safely as a dietary supplement, and we found it was selective for Kv1.1 over Kv1.2 (Fig. 4). We therefore pursued a fuller understanding of where gallic acid binds on Kv1.1 and how this might favor opening at more hyperpolarized membrane potentials. We first used SwissDock[47] for unbiased in silico docking of gallic acid to the AlphaFold-predicted[48,49] structure of human Kv1.1 (the high-resolution structure of which has not yet been reported). This predicted a cluster of binding poses on the Kv1.1 voltage sensing domain (VSD), close to Lys 195 (Fig. 10a). Interestingly, the location of the predicted VSD-binding site overlaps with a notable triple-residue sequence difference between Kv1.1 and Kv1.2 (Fig. 10b). Although the structure of Kv1.2 in complex with its cytoplasmic β subunit (Kvβ1.2) was previously solved using X-ray crystallography, the extracellular linkers of the VSD were only partially resolved in that structure[50]; similarly, the X-ray structure of the Kv2.1/Kv1.2 (KCNB1/Kv1.2) "paddle chimera" channel containing the Kv1.2 S1 to S3a region did not fully resolve the VSD extracellular linkers[51]. However, a more recent structure of the paddle chimera solved by cryo-electron microscopy resolved the linkers[52]. We therefore attempted docking in SwissDock of gallic acid to this structure but found that unlike in Kv1.1, gallic acid did not dock to the Kv1.2 S1 to S3a region (Fig. 10c). Neither did gallic acid dock to the VSD of the paddle chimera after the linker residues Kv1.2 Iso187 and Asn192 were changed to the Kv1.1 residues, glutamate and lysine, respectively (SwissDock did not permit switching of Kv1.2-Ser179 to isoleucine due to structural constraints) (Fig. 10d).

Based on the SwissDock prediction (Fig. 10a) we next used a K195-proximal gallic acid-binding pose (Fig. 10e) as the starting point for a 300-ns trajectory all-atom molecular dynamics (MD) simulation of gallic acid binding to the Kv1.1 VSD, to sample the accommodation of the VSD in the lipid bilayer and the interaction between the VSD and gallic acid. Figure 10f shows the residues present in the neighborhood

of gallic acid at least 50% of the time during the last 200 ns of the MD simulation. In the docking model, gallic acid is oriented parallel to the membrane surface, coordinated by R239 and E186, and appears to be forming a cation-pi like configuration with K195 (Fig. 10e). In the MD trajectory, gallic acid remains close to K195, but is oriented along the transmembrane direction and its carboxyl group forms persistent interactions with both R294 and K192, while the K195 center of charge is away from the ring (Fig. 10f; Supplementary Movie 1). We next ran a 300-ns trajectory, using the same initial gallic acid-bound configuration we used for the wild-type Kv1.1 VSD, for a mutant version of the Kv1.1 VSD in which the Kv1.1 residues highlighted in Fig. 10b were substituted with their Kv1.2 equivalents (Kv1.1-I182S,E190I,K195N, which we term Kv1.1-3 M). Unlike wild-type, in Kv1.1-3 M there was no development of a persistent bound configuration for gallic acid involving S4 and S1-S2 linker helix residues and gallic acid left the VSD (Supplementary Movie 2).

A comparison of the wild-type Kv1.1 VSD conformation with and without gallic acid bound reveals that the binding of gallic acid induces a slight reorientation of the S4-S3 paddle motif that leaves the first three S4 arginines (R236, R239, and R242) oriented towards the interior of the VSD (Fig. 10g). No such reconfiguration of the S4 arginines is found in Kv1.1.−3M (Fig. 10h). If the S4 arginines are not required to interact with the membrane surface in a VSD conformation consistent with an activated channel, this would be consistent with a reduction of the total electrical distance the S4 arginines must travel during activation.

We next pursued validation of the in silico predictions using functional and mutagenesis studies of the cluster of 3 VSD residues, given the Kv1.1 vs. Kv1.2 gallic acid selectivity (Fig. 4), the VSD docking data (Fig. 10a, e), and the MD simulation data (Fig. 10f).

We mutated all three VSD residues differentiating Kv1.1 from Kv1.2, creating Kv1.1-I182S, E192I, K195N (termed Kv1.1−3 M) and found that this modification substantially reduced sensitivity to gallic acid (Fig. 10g, h), such that maximal peak current augmentation at −40 mV was reduced fivefold compared to Kv1.1, and the $EC_{50}$ for augmentation was shifted from $379 ± 28$ nM gallic acid (wild-type Kv1.1) to not measurable because the effect was barely apparent even at $10^{-4}$ M gallic acid (Kv1.1−3 M) (Fig. 10i). Accordingly, the maximal gallic-induced hyperpolarization of $E_M$ in cells expressing Kv1.1-3 M was less than half that of oocytes expressing wild-type Kv1.1, and the $EC_{50}$ for $E_M$ hyperpolarization shifted from $18 ± 6$ nM (wild-type Kv1.1) to $345 ± 38$ nM (Kv1.1−3 M) (Fig. 10j). The functional data support a Kv1.1 gallic acid-binding site similar to that predicted by the MD simulation (Fig. 10f; Supplementary Movie 1).

## Discussion

Discovery of the basis for traditional Kwakwaka'wakw therapies for ataxia holds molecular mechanistic, cultural and translational significance. The location of the gallic acid-binding site, within the S1-S2 linker region on the extracellular side of the Kv1.1 voltage sensor, is unusual for a small molecule modifier of Kv channel activity. The Kv1 channel S1-S2 linker has been studied mostly because of the functional important glycosylation sites it contains, disruption of which isoform-dependently affects channel gating, surface trafficking and/or protein stability and surface expression of Kv1 channels[53–56], Recently, cryo-electron microscopy revealed that the Kv3.1 S1-S2 linker interacts with the S5-pore helix linker turret domain, strengthening coupling between the voltage sensor and the pore domain[57]. The Kv1.5 S1-S2 linker is responsive to mechanical stretch, leading to voltage-independent current augmentation[58]. Nature has exploited the S1-S2 linker as a binding site for peptide modulators of ion channel function; human beta-defensin-2, a cysteine-rich, low molecular weight peptide important in the innate immune response, inhibits the T-cell expressed Kv1.3 potassium channel by positive-shifting its voltage dependence of activation and slowing activation via a binding site that includes the

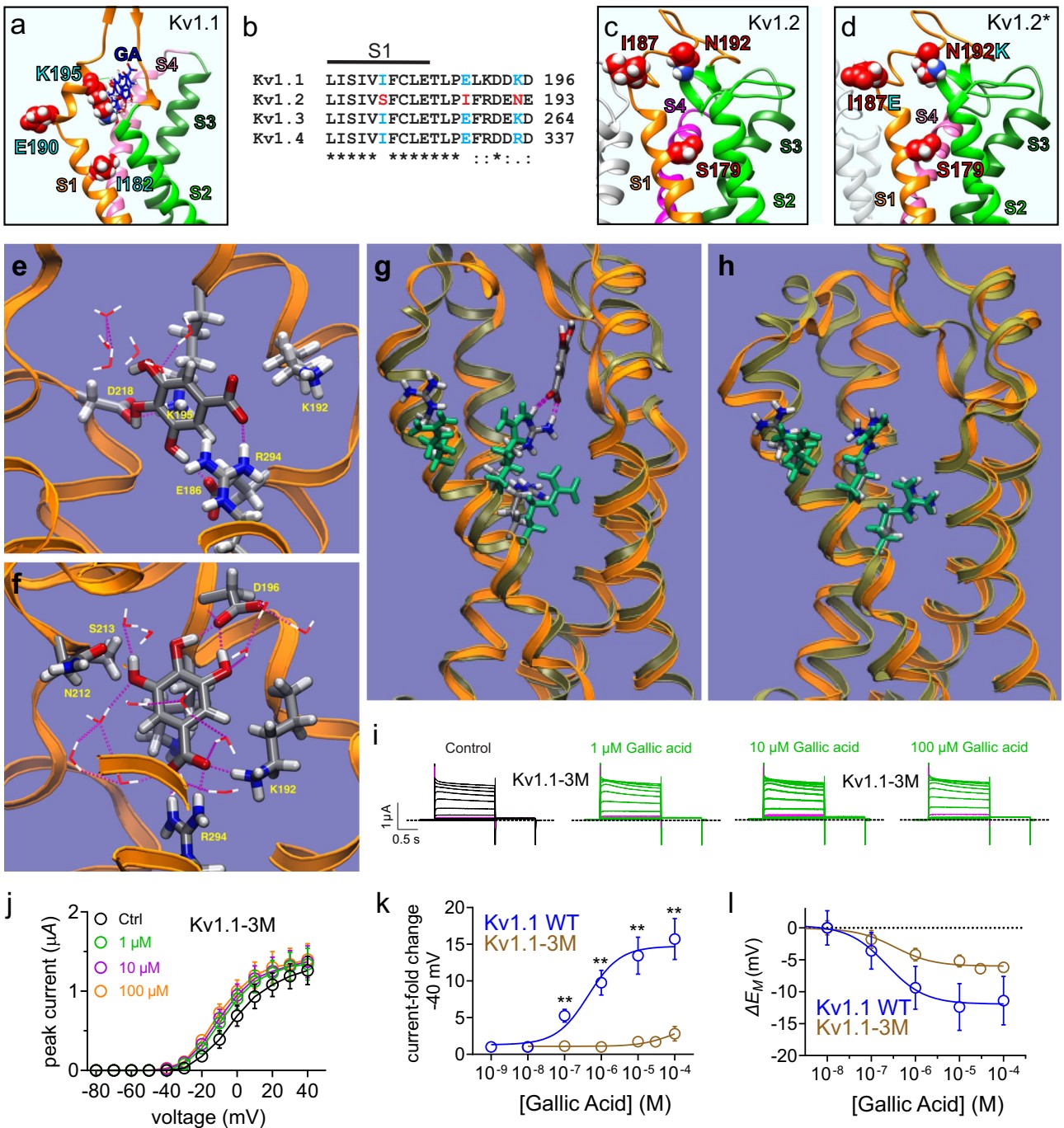

S1-S2 linker[59]. In addition, some scorpion toxins inhibit voltage-gated sodium channels by binding to the S1-S2 linker[60–63]. The S1-S2 linker is, however, relatively unexplored as a therapeutic small molecule-binding site, yet one that holds clear promise for beneficial modulation of Kv channel function, as we describe here for gallic acid based on our understanding of a traditional Native American ataxia therapy. Interestingly, our MD simulations predict that gallic acid binding to the S1-S2 linker reorientates the S4-S3 paddle motif such that the first three S4 arginines (R236, R239, and R242) now point towards the VSD interior (Fig. 10g). This supports a model in which gallic acid binding facilitates Kv1.1 activation (and rescues activity of EA1 mutant Kv1.1) by reducing the membrane electric field distance across which Kv1.1 S4 is required to travel for channel activation.

Native Americans have used at least 3000 plant species for medicine, and there may be other species for which knowledge of the medicinal use did not survive. The culture of many First Nations was disrupted or destroyed by European colonists and in addition, many First Nations did not use written documentation, instead using word of mouth to pass along records and traditions to subsequent generations. Documentation of Native America usage of plants to alleviate ataxia is unsurprisingly rare, given this and the relative rarity of ataxia compared to other maladies for which botanical therapies were mainstays, such as sore throat, skin and gastrointestinal disorders, and pain. Chapman Turner and Bell's 1973 ethnobotanical study of the Southern Kwakwaka'wakw First Nation nevertheless included documentation of use of three different plants for locomotor ataxia—bladderwrack kelp, common nettle and Pacific ninebark[32]. These three usages are the only mention of Native American folk medicine therapies for ataxia in the excellent Native American Ethnobotany database compiled by Moerman [http://naeb.brit.org/uses/]. These authors could not find

**Fig. 10 | Gallic acid binding to the Kv1.1 S1-S2 linker is predicted to reorientate S4 arginines towards the interior of the VSD.** Error bars indicate SEM. *n* indicates number of biologically independent oocytes. At least 2 batches of oocytes were used per experiment. Statistical comparisons by two-tailed paired *t*-test. Source data are available. **a** *Left*, SwissDock results for unbiased docking of gallic acid (GA; blue) to the AlphaFold-predicted structure of Kv1.1. VSD helices are individually labeled and colored. **b** Sequence alignment of human Kv1.1, 2, 3 and 4 partial S1 and S1-S2 linker with notable Kv1.2-specific sequence differences colored (cyan vs. red). *Residues are identical at this position in all sequences in the alignment; ":", conserved substitutions at this position in all sequences in the alignment; ".", semi-conserved substitutions at this position in all sequences in the alignment. Arrows, S1/2 linker residues predicted by docking and/or MD simulations to interact with gallic acid. **c** SwissDock unbiased docking of gallic acid to Kv1.2 produced no GA docking to the VSD. Notable Kv1.2-specific sequence differences from Panel B are colored red and labeled. **d** SwissDock unbiased docking of gallic acid to Kv1.2 with mutations made to S1-S2 linker residues as indicated in cyan to match Kv1.1 residues (Kv1.2*) produced no GA docking to the VSD. **e** Starting configuration for MD simulations of gallic acid (red) binding to the Kv1.1 VSD structure predicted by Alphafold. Residues interacting with gallic acid are shown in licorice representation colored by atom (C, silver; N, blue; O, red; H, white). Waters in the solvation shell of gallic acid are rendered as lines colored by atom. Dashed lines, H-bond like configurations. **f** Predominant localization of gallic acid for the last 200 ns of the 300-ns MD trajectory. Residues present in the neighborhood of gallic acid at least 50% of the time the last 200 ns of the MD trajectory are shown in licorice representation colored by atom. Waters in the solvation shell of gallic acid are rendered as lines colored by atom. Dashed lines, H-bond like configurations. **g** Superimposed configuration snapshots from the end of the wild-type Kv1.1 simulations with and without gallic acid. The VSD backbone is shown in ribbon representation (orange, with gallic acid bound; tan, without gallic acid). For the gallic acid-bound simulation system, the first three S4 arginines (R236, R239, and R242) and gallic acid are shown in licorice representation colored by atom (C, silver; N, blue; O, red; H, white). The S4 arginines in the simulation system without gallic acid are colored green. **h** A similar superimposed configuration as in panel **g**, shown for the Kv1.1−3 M simulation system. **i** Mean traces for Kv1.1-I182S,E192I,K195N (termed Kv1.1−3 M) channels expressed in oocytes in the absence (Control) or presence of gallic acid (1–100 μM). Scale bars central; voltage protocol as in Fig. 2; *n* = 5 per group. Magenta traces show same-voltage pulses in each pair for comparison. **j** Mean peak current (measured during prepulse) from traces as in **i**; *n* = 5 per group. **k** Mean peak current fold-change vs. [gallic acid] for A1-3M traces in **i** and wild-type Kv1.1; 0.1 μM gallic acid (*n* = 5; *p* = 0.0085); 1 μM gallic acid (*n* = 5; *p* = 0.0066); 10 μM gallic acid (*n* = 5; *p* = 0.0085); 100 μM gallic acid (*n* = 5; *p* = 0.0074). **l** Mean change in unclamped oocyte membrane potential (Δ$E_M$) vs. [gallic acid] for oocytes expressing A1-3M (as in **i**) or wild-type Kv1.1; *n* = 5.

evidence in the literature suggesting that ataxia was particularly prevalent in the Kwakwaka'wakw vs. other populations, suggesting that the scarcity of documented Native American folk ataxia therapies for other populations is a documentation bias (i.e., lack of records for other First Nations for this disorder) rather than reflective of a particular epidemiological phenomenon. Comparing remedies for paralysis, for example, there are >30 in Moerman's Native American Ethnobotany database, attributed to 18 different First Nations.

The Kwakwaka'wakw are the earliest known, and likely original, inhabitants of what is now referred to as northern Vancouver Island in British Columbia, the adjacent mainland, and also the other islands between the two[64]. The Kwakwaka'wakw passed down their stories and traditions via "potlatch" ceremonies, which were outlawed in 1884 by the Canadian government to attempt to destroy Kwakwaka'wakw traditions[65]. Undeterred, the Kwakwaka'wakw continued potlatch ceremonies in secret, despite arrests and convictions, until the potlatch ban was overturned in 1951, followed later by returning of ceremonial treasures confiscated during the ban. Fortunately, many Kwakwaka'wakw traditions, including their ingenious ataxia therapies, survived the period of cultural genocide and were subsequently documented in writing[64].

The Kwakwaka'wakw treated locomotor ataxia by various methods. They did not consume bladderwrack kelp but instead rubbed fresh kelp on the legs and feet of those with ataxia[66]. Whether or not this could deliver sufficient gallic acid to the peripheral nerves where it might beneficially modulate Kv1 subfamily channels is open to debate, but it is relevant to note that, e.g., following massage with lavender oil, linalool (the most common terpene present in lavender) and linalyl acetate can be detected in the blood rapidly, reaching peak levels in under 20 min and subsiding to low levels after 90 min[67]. The Kwakwaka'wakw did, however, ingest Pacific ninebark root extract, although its use for ataxia is described as a purgative during recovery from locomotor ataxia[66]. Perhaps most strikingly, Kwakwaka'wakw used nettles to treat ataxia by first cutting the soles of the feet with shells and then rubbing them with nettles (and hellebore roots)[66]. One can imagine that this would be a highly effective method for introducing nettle phenolics such as gallic acid and hydrolysable tannins directly into the bloodstream, helping to ensure maximal bioavailability while avoiding the gastrointestinal side effects of tannic acid[68].

Given the relative rarity of ataxia in modern populations globally, it is perhaps surprising that the Kwakwaka'wakw had three different treatments for it. There are an estimated 25,000 people in Canada, and 150,000 people in the United States, currently with some form of acquired or inherited ataxia—and an estimated incidence globally of 0.3–41/100,000 depending on the region[69,70]. The Kwakwaka'wakw population is thought to have peaked at about 20,000 prior to European contact[71], and in the 2016 census those self-identifying as having Kwakwaka'wakw ancestry numbered just 3,665 people[72]. In such a small population even at its peak, ataxia must have been at exceedingly low incidence, yet this is based on numbers for modern populations, and there may have been historical causes of acquired ataxia that are less prevalent now. These might include acute injury or infection, vitamin deficiency, or perhaps ingestion of toxic plants, all of which are and were likely more common in populations living off the land, hunting and lacking access to modern antibiotics and supplements. As an example, ingestion of the hallucinogenic plants in the *Datura* genus, such as *D. stramonium* (Jimson Weed, Locoweed, Devil's Trumpet) revered in indigenous medicine and shamanism, can cause ataxia if improperly dosed[73].

EA1 would have been even less frequent, and so the molecular basis for gallic and tannic acid modulation of Kv1.1 to ameliorate ataxia would have relied upon beneficial modulation of wild-type Kv1.1 to treat ataxia for which the pathologic basis was not necessarily Kv1.1 dysfunction. It is not too much of a leap to imagine the dampening in excitability in peripheral nerves induced by augmenting Kv1.1 (and Kv1.2) activity being beneficial in forms of ataxia not necessarily linked to Kv1.1 dysfunction. Retigabine, the anticonvulsant that works by negative-shifting the voltage dependence of KCNQ2/3 channel activation, is effective in forms of epilepsy not caused by KCNQ2/3 channel dysfunction, as well as those that are[74]. Similarly, *Mallotus oppositifolius* extract, which also augments KCNQ2/3 channel activity (via isovaleric acid and mallotoxin), is used in African folk medicine to treat seizures not necessarily caused by KCNQ2/3 dysfunction;[25] the same applies to (E)-2-dodecenal, found in cilantro and culantro[26]. Kv1 subfamily channels influence neuronal excitability, action potential threshold and synaptic transmission, and Kv1 inhibitors induce somatic depolarization and increase spontaneous firing rates, for example in cerebellar neurons[12]. Thus, much like retigabine activation of KCNQ2/3, Kv1 current augmentation by small molecules such as gallic acid would be expected to dampen excess neural excitability and potentially ameliorate ataxia even if Kv1 channel dysfunction was not the original cause.

The discovery of a binding site for gallic acid in an unexpected location within the S1-S2 linker at the extracellular face of the VSD on Kv1.1 presents therapeutic possibilities for ataxia and other Kv1.1-linked disorders, including epilepsy. Gallic acid is already a

widely used supplement, with claimed beneficial effects as an anti-oxidant, antimicrobial, anti-obesity and anti-cancer compound that may also improve brain function[75]. Our finding that gallic acid can reverse the loss of function caused by some EA1-linked *KCNA1* sequence variants has the potential for immediate clinical significance. Gallic acid is a well-tolerated supplement, the use of which does not require FDA approval. Therefore, clinicians treating EA1 patients, especially those whose ataxia is intractable to current standard treatments such as carbamazepine, or whose patients do not tolerate standard treatments well, have the option of trying gallic acid supplementation, or even, for example, kelp extract, which is also readily available over the counter. Quantitative, controlled clinical trials are warranted to investigate whether the ability of the extracts described herein, and also their component gallic acid, to rescue some function of selected EA1 Kv1.1 mutants (and enhance function of wild-type Kv1.1 and Kv1.2 depending on the extract), will translate into in vivo benefits.

Tannic acid was also effective but is considered less of a viable drug or lead compound because of gastric side effects[68]. Moreover, given the structural simplicity of gallic acid, medicinal chemistry modifications to improve selectivity, efficacy and potency while still targeting the Kv1.1 VSD-binding site may lead to advances in ataxia therapy, which currently relies on repurposing epilepsy drugs such as carbamazepine, due to the previous lack of a potent and selective drug to restore mutant Kv1.1 function[76]. We previously found that in KCNQ2/3 channels, gallic acid binds instead in the established retigabine/GABA-binding site bounded by an S5 Trp residue (W265 in human KCNQ3) and an S4-5 linker Arg residue (R242 in human KCNQ3)[77], suggesting that the small, simple structure of gallic acid facilitates pharmacological versatility that for specificity reasons may need to be overcome by medicinal chemistry. On the other hand, mild activation of KCNQ2/3 would likely be more beneficial than harmful in the context of EA1. Similarly, we found that some extracts also activated Kv1.2 (Fig. 1) and further in vivo testing will be important to decide whether this is beneficial or problematic in various ataxia models; it is also important to note that *KCNA2* gene variants are also implicated in forms of episodic ataxia associated with myoclonic epilepsy, spastic paraplegia, and also epileptic encephalopathy in the absence of ataxia[78–80]. In future work, we will examine the efficacy of the extracts and their constituents in counteracting the effects of ataxia-linked *KCNA2* variants.

It is essential to recognize that the discovery of a molecular mechanistic approach to augment Kv1.1 channel function with therapeutic potential was facilitated by the inventiveness and ethnobotanical expertise of the Kwakwaka'wakw people. Without preservation of the ethnobotanical practices of indigenous peoples and the environments that sustain them, we stand to lose a vast natural apothecary and the means to therapeutically leverage it.

## Methods

### Preparation of plant extracts
We purchased live *Urtica dioica* (Common Nettle) from Crimson Sage Nursery (Orleans, CA, USA), cultivated it in Irvine CA, collected the aerial portions (primarily leaves) and then froze them until the day of extraction. We purchased *Physocarpus capitatus* (Pacific ninebark) live whole plant from Ozark Wildflower Company (Jasper, AR, USA) and separated the roots and aerial parts before extraction of each. We homogenized plant samples using a bead mill with porcelain beads in batches in 50 ml tubes (Omni International, Kennesaw, GA, United States). We purchased certified organic tree bark powders from Mountain Rose Herbs (Eugene, OR, USA) and dry raw *Sophora japonica* (fine pellets) from Teriya (People's Republic of China). We purchased bladderwrack kelp extract capsules from Terra Vita (Brampton, Ontario, Canada) and discarded the outer capsule before extraction. We resuspended all plant powders and homogenates in

80% methanol/20% water (100 ml per 5 g solid) and then incubated for 48 h at room temperature, occasionally inverting the bottles to resuspend the extracts. We next filtered the extracts through Whatman filter paper #1 (Whatman, Maidstone, UK), and then removed the methanol using evaporation in a fume hood for 24–48 h at room temperature. We next centrifuged extracts for 10 min at 15 °C, 4000 RCF to remove the remaining particulate matter, followed by storage at −20 °C. On the day of electrophysiological recording, we thawed the extracts and diluted them 1:50 in bath solution (see below), to equal 5 mg plant solid starting material per ml, immediately before use.

### Channel subunit cRNA preparation and *Xenopus laevis* oocyte injection
Wild-type and mutant Kv1 cDNAs were generated (Genscript, Piscataway, NJ) in the pTLNx expression vector. As previously described[24], we generated cRNA transcripts encoding human Kv1.1 (wild-type and mutant) and Kv1.2, by in vitro transcription using the mMessage mMachine kit (Thermo Fisher Scientific), after vector linearization, from cDNA sub-cloned into plasmids (pTLNx) incorporating *Xenopus laevis* β-globin 5′ and 3′ UTRs flanking the coding region to enhance translation and cRNA stability. We injected defolliculated stage V and VI *Xenopus laevis* oocytes (Xenoocyte, Dexter, MI, US) with Kv1 cRNAs (1–5 ng). We incubated the oocytes at 16 °C in ND96 oocyte storage solution containing penicillin and streptomycin, with daily washing, for 2 days prior to two-electrode voltage-clamp (TEVC) recording.

### Two-electrode voltage clamp (TEVC)
We performed TEVC at room temperature using an OC-725C amplifier (Warner Instruments, Hamden, CT) and pClamp10 software (Molecular Devices, Sunnyvale, CA) 2 days after cRNA injection as described in the section above. For recording, oocytes we placed in a small-volume oocyte bath (Warner) and viewed with a dissection microscope. We sourced chemicals from Sigma. We studied effects of plant extracts and of compounds, solubilized directly in bath solution (in mM): 96 NaCl, 4 KCl, 1 MgCl₂, 1 CaCl₂, 10 HEPES (pH 7.6). We introduced extracts or compounds into the oocyte recording bath by gravity perfusion at a constant flow of 1 ml per minute for 3 min prior to recording. Pipettes were of 1–2 MΩ resistance when filled with 3 M KCl. We recorded currents in response to voltage pulses between −80 mV and +40 mV at 10 or 20 mV intervals from a holding potential of −80 mV, to yield current-voltage relationships and examine activation kinetics. To calculate the voltage dependence of activation ($V_{0.5}$), tail currents were recorded at a voltage pulse of −50 mV immediately following prepulse voltages between −80 mV and +40 mV. We analyzed data using Clampfit (Molecular Devices) and Graphpad Prism software (GraphPad, San Diego, CA, USA), stating values as mean ± SEM. We plotted raw or normalized tail currents vs. prepulse voltage and fitted with a single Boltzmann function:

$$g = \frac{(A_1 - A_2)}{\left\{1 + \exp[V_{\frac{1}{2}} - V/Vs]\right\} y + A_2} \tag{1}$$

where $g$ is the normalized tail conductance, $A_1$ is the initial value at -∞, $A_2$ is the final value at +∞, $V_{1/2}$ is the half-maximal voltage of activation and $V_s$ the slope factor.

### Analytical chemistry
Gallic acid was obtained from Sigma-Aldrich. Methanol (MeOH) was HPLC or LC/MS grade from Fisher or VWR International. Water was 18.2 mΩ·cm from a Barnstead NANOpure Diamond™ system. Trifluoroacetic acid was obtained from EMD Millipore or Alfa-Aesar. Mass spectroscopy employed a Thermo Scientific TSQ Quantum Ultra triple

stage quadrupole mass spectrometer. Heated-electrospray ionization (H-ESI) was used in negative or positive ionization mode depending on the structure of the analyte. Automatic methods for the optimization of instrument parameters were used to maximize sensitivity. Samples were analyzed by direct injection in MeOH or MeOH/water (TFA conc kept at 0.01% or less) using a syringe pump. Gallic acid was identified from molecular ion $m/z$ 169 (M-H$^+$) in negative ionization mode and by daughter ion analysis ($m/z$ 125, collision energy 10). Preparative reversed phase high-performance liquid chromatography (RPHPLC) separations were carried out using a Shimadzu system consisting of two LC-8A pumps, a fraction collector (FRC-10A), a SIL-10AP auto sampler, a diode array detector (CPD-M20A) and a CBM-20A communication module. The separations employed a Waters PREP Nova-Pak® HR C18 6 μM 60 Å 40 × 100 mm reversed phase column with a 40 × 10 mm Guard-Pak insert and a Waters PrepLC Universal Base. The solvent system employed was a MeOH/water gradient (15 to 100% MeOH over 5 min) both containing 0.1 % TFA. Fractions were collected based on their response at 254 nm.

### In silico docking

For in silico ligand docking predictions of binding to Kv1 channels, we performed unguided docking to predict potential binding sites, using SwissDock with CHARMM forcefields[47,81] and channel PDB files as described in the text. We used AlphaFold[48,49] to generate a predicted structure for the Kv1.1 VSD. We prepared channel structures for docking using DockPrep in UCSF Chimera https://www.rbvi.ucsf.edu/chimera)[82], with which we also generated docking figures.

### Molecular dynamics

We performed MD simulations of a single human Kv1.1 VSD (residues 148 through 307) and Kv1.1–3 M VSD with and without gallic acid bound embedded in a POPC bilayer in 150 mM NaCl. The initial VSD configuration for the Kv1.1 VSD with gallic acid bound was the one resulting from the AlphaFold[48,49] and SwissDock[47,81] predictions. The full simulation system initial configuration was generated using Membrane Building in CHARMM-GUI[83,84] and the VMD 1.9.4 software package[85] with the positioning of the VSD in the membrane as predicted by the PPM server[86]. The final simulation system consisted of 19,844 waters, 271 lipids, 107 ions, 1 gallic acid molecule and 1 VSD, for a total of 98,624 atoms. For the other three simulations we used the same overall initial configuration after removing the gallic acid molecule and/or performing the three-residue sidechain substitution.

The simulations were performed using the NAMD3 software package[87]. Each simulation system was subjected to 1000 steps of conjugate gradient energy minimization followed by a 100-ps run at constant temperature (300 K) and constant volume with harmonic restraints on the protein heavy atoms, the gallic acid, and the lipid headgroups. The harmonic restraints were progressively released over five 200-ps runs at constant temperature (310 K) and pressure (1 atm). We ran each simulation for 300 ns. The CHARMM36m[88] and CHARMM36[89] force fields were used to model the protein and lipids, respectively. The TIP3P model was used for water[90]. Topology and parameters for gallic acid were generated using SwissParam[91]. The smooth particle-mesh Ewald summation method[92,93] was employed for the calculation of electrostatic interactions. Short-range real-space interactions were cut off at 12 Å, employing a force-based switching function. A reversible time step algorithm[94] was used to integrate the equations of motion with a time step of 4 fs for electrostatic forces, 2 fs for short-range non-bonded forces and bonded forces. All bond lengths involving hydrogen atoms were held fixed using the SHAKE algorithm[95]. A Langevin dynamics scheme was used for temperature control and a Nosé–Hoover–Langevin piston was used for pressure control[96,97].

### Statistics and reproducibility

All values are expressed as mean ± SEM. Multiple comparison statistics were conducted using a one-way ANOVA with a post hoc Tukey HSD. Comparison of two groups was conducted using a $t$-test; all $p$-values were two-sided.

### Reporting summary

Further information on research design is available in the Nature Portfolio Reporting Summary linked to this article.

## Data availability

The data that support this study are available from the corresponding authors upon request. Source data for Figs. 1–10, and the initial and final configurations for the MD simulations, are available at Dryad [https://doi.org/10.7280/D1569B]. Source data are provided with this paper.

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

## Acknowledgements

This study was supported by the National Institutes of Health, National Institute of General Medical Sciences (GM130377) and a Samueli Scholarship (Susan Samueli Integrative Health Institute, UC Irvine) to GWA. We are grateful to Dr. Derk Hogenkamp for gallic acid analyses.

## Author contributions

G.W.A. and R.W.M. conceived the study; R.W.M. conducted T.E.V.C. analysis and analyzed data; R.S. contributed clinical information and advice; J.A.F. prepared and performed MD simulations; J.A.F. and D.J.T. interpreted MD data; G.W.A. oversaw and obtained funding for the project; R.W.M., J.A.F., and G.W.A. prepared the figures, G.W.A. wrote the manuscript; all authors contributed to editing the manuscript.

## Competing interests

The authors declare no competing interests.
