## [Peer Review File · Nature Communications]

Native American ataxia medicines rescue ataxia-linked mutant potassium channel activity via binding to the voltage sensing domainReviewers' Comments:

Reviewer #1:

Remarks to the Author:

This manuscript by Manville et al used two-electrode voltage clamp recordings in oocytes to identify surprising and novel pharmacological effects of native American remedies for augmenting function of mutant potassium channels associated with ataxia. The paper begins by screening previously overlooked native American plant extracts ataxia treatments for activity on potassium channel currents, specifically on voltage-gated Kv1.1 and Kv1.2 subunits. KCNA1 (Kv1.1) is known to be a genetic cause of episodic ataxia, whereas Kv1.2 is a closely related isoform investigated as a measure of specificity. These screens identified several extracts that had effects on these potassium channels. Therefore, they investigated compounds known to be in these plant extracts leading to identification of gallic and tannic acids as potential active ingredients mediating the actions. They then tested the action of these acids on WT Kv1.1 and known mutant variants of Kv1.1 associated with ataxia. They found that gallic acid especially can act as a specific opener of WT Kv1.1 subunits with the ability to improve function of some mutant Kv1.1 channel variants. Finally, using *in silico* modeling, the authors identified a potential binding site for gallic acid in the voltage sensing domain of Kv1.1, which they tested functionally in cells. Overall, this was a very interesting study with potentially important clinical implications for identifying new therapeutic approaches for treatment of episodic ataxia and Kv1.1 channelopathy. That the novel reported pharmacological agents were identified from native American records of disease treatments adds further to the novelty and interest of the study. Despite these strengths, the paper had several weaknesses (summarized below) that should be addressed because they detracted from the potential significance and rigor of the study. In general, these weaknesses centered around imprecise nomenclature and data reporting, which lessened the clarity of the manuscript. In addition, the overall translational significance of the study's findings was not made clear in the manuscript, especially in the Discussion section.

MAJOR CONCERNS:

1) That historical native American ataxia remedies can improve function of Kv1.1 channels known to be important in episodic ataxia is of great clinical and cultural interest. However, the potential significance of the reported findings is barely mentioned in the Discussion section, which is instead reserved for lengthy discussion of historical and cultural contexts. Although future directions are discussed, the Discussion section contains no clear summary of the paper's main conclusions or significant contributions to the field. In addition, the potential translational limitation of the study in analyzing Kv1.1-only containing tetramers is not discussed (as discussed in 2b-c below, *in vivo* channel tetramers contain heterotetrameric stoichiometry with Kv1.1 and other subunits)

2) Several issues lessen the clarity and context of the paper:

a. The authors deliberately chose to refer to the protein under examination as KCNA1 throughout the paper (line 60) but this is unnecessarily confusing and unorthodox since the ion channel field almost always refers to the protein as Kv1.1.

b. It is stated that Kv1.1 forms homomeric channels in axons (line 73) but to my knowledge there are no known reported examples of this in the literature. The two refs #7 and #8 cited for this assertion only examine homomeric channels that have been generated *in vitro*, not *in vivo* in actual neurons. This stoichiometry discrepancy could have major effects on pharmacology.

c. Related to the previous comment, the channel subunit composition is also referred to imprecisely in lines 268-270, when it is stated that EA1 patients typically present with "heteromeric mutant channels." It appears that the authors are confusing the terms "heteromeric" and "heterozygous." Almost all EA1 patients are heterozygous for KCNA1 mutations. But modeling this heterozygosity with "one wild-type and one mutant allele" (line 269) does not mimic this condition because, as mentioned in 2b above, channels are composed of heterotetramers, meaning a Kv1.1 subunit plus some other Kv1-family subunits such as Kv1.2 or Kv1.4.

d. Stating the extract concentrations as a dilution ratio (e.g., in line 127) seems not very precise since the amount of extract obtained in each preparation could differ between samples. Is it possible to express this in a more quantifiable way?

e. Sometimes the language used to refer to the currents measured is ambiguous because it is not stated whether it is peak currents or tail currents being referenced. For example, in line 147-148, it only says "increasing current across the voltage range tested" without specifying which type of current was being examined.

f. After identifying pharmacological effects of plant extracts in the first section of the Results, the next section of the Results begins to screen compounds present in these extracts settling on gallic and tannic acids as some of the most effective. However, it is never clearly stated which of the plant extracts have these two compounds (lines 154-157), just saying "some of which contain one or both."

g. There are many incongruencies between the figures and figure legends and the text. For example, Figures 1 and 5 have panels A-D but the legends report A-E so some data is missing in the figures. Sometimes the text reports data not shown in figures, such as for birch bark (line 159) which does not appear in Fig 3 as stated. In other cases, graph axes are mislabeled such as in Fig 6D where the first data point is for -30mV but the text reports it as -40mV (line 228). In line 135, it states that the slope of KCNA2 voltage dependence was steepened 2-fold by kelp extract but this is not evident in Figure 1C or Supplemental Figure 1, bringing up the question of whether the result was misstated or whether that figure panel is missing.

h. In the exploration of dose response to gallic and tannic acids and rutin, why are current fold changes measured at different holding potentials making the results less comparable for each compound?

i. In the in silico modeling, why is the binding site of gallic acid not explicitly stated? Instead it is described as "overlapping with" or binding "close to" (line 322-323) the voltage sensing domain and associated residues. It would be helpful to highlight if possible where exactly the binding site is with relation to the sequences shown in Fig 10B. Fig 10B also has some problems with the legend as the meaning of the various symbols used are never described (eg, the * and : symbols).

3) Some corrections are needed with the methods described. The in silico docking experiments are stated as having been used on KCNQ channels (line 544) which is surely a typo since this study deals with Kv1.1. In addition, the statistics description is lacking. It states 1-way ANOVA was used throughout (line 552-553) but there are many figure panels that compare membrane potential for two variables, which most likely used t-tests instead.

4) The title of the paper is misleading when it says the medicines rescue "via a novel voltage sensor binding site," which makes it sound like a new voltage sensor site was identified in Kv1.1. Instead, from the description in the text, the voltage sensor site itself was not newly identified but rather it is a novel site for gallic acid to bind. Maybe a better more accurate phrase would be that the medicines rescue "via binding to the voltage sensor domain."

Reviewer #2:

Remarks to the Author:

This is a very interesting study inspired by ethnobotanical knowledge of some useful herbs in treating or helping with Episodic Ataxia 1. The authors followed rational steps first to validate the beneficial effects of some herbal extracts and then identify the major bioactive constituents mediating the

desired enhancement of the EA1-relevant KCNA1 mutants. The manuscript is very well written and thoroughly enjoyable and the findings will be attractive to readers from a diverse field including ion channels, ethnopharmacology/alternative medicine, drug discovery, neuroscience etc.

I have the following points:

Major point:

Whilst it is great that the authors could identify potential binding sites via which Gallic acid was acting, it would have been great to have some thoughts on possible molecular mechanism of why binding to such region could enhance KCNA1 current. Also it is commendable that they have used an unbiased docking approach and experimentally validated key residues, nevertheless the docking was done against a static structure of the channel protein. I recommend that the authors consider:

- MD simulation to evaluate how stable the docked pose of gallic acid was.
- MD simulation of the unliganded and docked KCNA1 mutant proteins and comparative analyses of the simulation trajectories which could potentially highlight the molecular mechanism of gallic acid's action in this regard.

Minor point:

Figure 1

Plant names should be italicized

In several figures, it has been mentioned that 'n indicates number of oocytes'. It should be ideal to additionally specify the number of independent days of experiments performed. Also it has been mentioned that 'statistical comparisons by one way ANOVA' - it was not clear what post hoc test was used.

My co-authors and I are very grateful for the careful reviews and positive response to our manuscript. We have conducted additional electrophysiological and analytical chemistry studies to address comments from Reviewer 1 and collaborated with Molecular Dynamics experts to produce simulations to address comments from Reviewers 1 and 2. We have also thoroughly revised the manuscript and figures. New experiments and edits in response to reviewer comments are described point-by-point below in bold, with reviewers' questions shown in italics. In the manuscript itself, changes are highlighted in yellow.

REVIEWER COMMENTS

Reviewer #1 (Remarks to the Author):

This manuscript by Manville et al used two-electrode voltage clamp recordings in oocytes to identify surprising and novel pharmacological effects of native American remedies for augmenting function of mutant potassium channels associated with ataxia. The paper begins by screening previously overlooked native American plant extracts ataxia treatments for activity on potassium channel currents, specifically on voltage-gated Kv1.1 and Kv1.2 subunits. KCNA1 (Kv1.1) is known to be a genetic cause of episodic ataxia, whereas Kv1.2 is a closely related isoform investigated as a measure of specificity. These screens identified several extracts that had effects on these potassium channels. Therefore, they investigated compounds known to be in these plant extracts leading to identification of gallic and tannic acids as potential active ingredients mediating the actions. They then tested the action of these acids on WT Kv1.1 and known mutant variants of Kv1.1 associated with ataxia. They found that gallic acid especially can act as a specific opener of WT Kv1.1 subunits with the ability to improve function of some mutant Kv1.1 channel variants. Finally, using in silico modeling, the authors identified a potential binding site for gallic acid in the voltage sensing domain of Kv1.1, which they tested functionally in cells. Overall, this was a very interesting study with potentially important clinical implications for identifying new therapeutic approaches for treatment of episodic ataxia and Kv1.1 channelopathy. That the novel reported pharmacological agents were identified from native American records of disease treatments adds further to the novelty and interest of the study. Despite these strengths, the paper had several weaknesses (summarized below) that should be addressed because they detracted from the potential significance and rigor of the study. In general, these weaknesses centered around imprecise nomenclature and data reporting, which lessened the clarity of the manuscript. In addition, the overall translational significance of the study's findings was not made clear in the manuscript, especially in the Discussion section.

MAJOR CONCERNS:

1) That historical native American ataxia remedies can improve function of Kv1.1 channels known to be important in episodic ataxia is of great clinical and cultural interest. However, the potential significance of the reported findings is barely mentioned in the Discussion section, which is instead reserved for lengthy discussion of historical and cultural contexts. Although future directions are discussed, the Discussion section contains no clear summary of the paper's main conclusions or significant contributions to the field. In addition, the potential translational limitation of the study in analyzing Kv1.1-only containing tetramers is not discussed (as discussed in 2b-c below, in vivo channel tetramers contain

heterotetrameric stoichiometry with Kv1.1 and other subunits)

- **We have added discussion of the significance, as requested, to the Discussion section (pages 16, 17, 21). See below for response regarding heterotetramers.**

2) *Several issues lessen the clarity and context of the paper:*

a. The authors deliberately chose to refer to the protein under examination as KCNA1 throughout the paper (line 60) but this is unnecessarily confusing and unorthodox since the ion channel field almost always refers to the protein as Kv1.1.

- we have switched to the Kv1.1 and Kv1.2 nomenclature throughout when referring to the protein

b. It is stated that Kv1.1 forms homomeric channels in axons (line 73) but to my knowledge there are no known reported examples of this in the literature. The two refs #7 and #8 cited for this assertion only examine homomeric channels that have been generated in vitro, not in vivo in actual neurons. This stoichiometry discrepancy could have major effects on pharmacology.

- thank you for pointing this out - we have clarified this and conducted additional experiments, on heteromers (see response to next point)

c. Related to the previous comment, the channel subunit composition is also referred to imprecisely in lines 268-270, when it is stated that EA1 patients typically present with “heteromeric mutant channels.” It appears that the authors are confusing the terms “heteromeric” and “heterozygous.” Almost all EA1 patients are heterozygous for KCNA1 mutations. But modeling this heterozygosity with “one wild-type and one mutant allele” (line 269) does not mimic this condition because, as mentioned in 2b above, channels are composed of heterotetramers, meaning a Kv1.1 subunit plus some other Kv1-family subunits such as Kv1.2 or Kv1.4.

- we previously used “heteromeric” to refer to mixed wild-type/mutant channels, as “heterozygous” generally refers to genes/alleles, not proteins. However, we agree that in a system that involves mutations as well as channel isoform heteromers, our terminology was confusing. We have now clarified the nomenclature we use to describe the various channel types and use “heterozygous” when referring to the genetic description of patients (page 12, line 305) and only use “heteromeric” when referring to mixed-Kv1 isoform channels (page 12). When we are studying channels made from both wild-type and mutant subunits, we spell that out. Importantly, we have also conducted additional experiments using co-injection of wild-type Kv1.1, mutant Kv1.1 and wild-type Kv1.2 cRNA and studied the response to gallic acid of heteromeric Kv1.1/Kv1.2 channels containing a Kv1.1 mutation. We show that at concentrations as low as 1 μ M, gallic acid can restore function of these channels (Supplementary Figures 12 and 13).

d. Stating the extract concentrations as a dilution ratio (e.g., in line 127) seems not very precise since the amount of extract obtained in each preparation could differ between samples. Is it possible to express this in a more quantifiable way?

- we now express as a function of the mass of plant matter used in the extractions (“1:50 dilution” = extract from 5 mg of plant solid starting material per ml of bath solution)

e. Sometimes the language used to refer to the currents measured is ambiguous because it is not stated whether it is peak currents or tail currents being referenced. For example, in line 147-148, it only says “increasing current across the voltage range tested” without specifying which type of current was being examined.

- we have clarified this throughout

f. After identifying pharmacological effects of plant extracts in the first section of the Results, the next section of the Results begins to screen compounds present in these extracts settling on gallic and tannic acids as some of the most effective. However, it is never clearly stated which of the plant extracts have these two compounds (lines 154-157), just saying “some of which contain one or both.”

- we have clarified this (lines 155-163) and we conducted additional experiments to confirm presence of gallic acid in Kelp and Pacific ninebark extracts (lines 163-166; Supplementary Figure 1).

g. There are many incongruencies between the figures and figure legends and the text. For example, Figures 1 and 5 have panels A-D but the legends report A-E so some data is missing in the figures. Sometimes the text reports data not shown in figures, such as for birch bark (line 159) which does not appear in Fig 3 as stated. In other cases, graph axes are mislabeled such as in Fig 6D where the first data point is for -30mV but the text reports it as -40mV (line 228). In line 135, it states that the slope of KCNA2 voltage dependence was steepened 2-fold by kelp extract but this is not evident in Figure 1C or Supplemental Figure 1, bringing up the question of whether the result was misstated or whether that figure panel is missing.

- thank you for pointing these out - we have reviewed all the figures and legends for accuracy and corrected where needed. We incorrectly stated that the Kv1.2 slope was altered; in fact, it was solely the Kv1.1 slope that was altered, as apparent in the Figure 1 C graph and also in the Supplementary Data tables for Figure 1. We now also state the slope values in the text (lines 139-141).

h. In the exploration of dose response to gallic and tannic acids and rutin, why are current fold changes measured at different holding potentials making the results less comparable for each compound?

- the potentials at which each compound exerts its greatest effects differ between compounds. To address the concern we now plot shift in midpoint voltage dependence of activation, instead of fold change at a specific voltage, versus [compound], to permit direct comparison of their effects (new Figure 4).

*i. In the in silico modeling, why is the binding site of gallic acid not explicitly stated? Instead it is described as “overlapping with” or binding “close to” (line 322-323) the voltage sensing domain and associated residues. It would be helpful to highlight if possible where exactly the binding site is with relation to the sequences shown in Fig 10B. Fig 10B also has some problems with the legend as the meaning of the various symbols used are never described (eg, the * and : symbols).*

- in response to Reviewer 2 we have now performed Molecular Dynamics simulations to improve the binding site prediction and in response to Reviewer 1 we have clarified the data presentation. We include the new data in the revised Figure 10 and new Supplementary Videos 1 and 2, and describe the interactions between gallic acid and specific residues in the VSD more clearly in the Results, page 15. We have also added a key to the symbols in the figure legend as requested.

3) Some corrections are needed with the methods described. The *in silico* docking experiments are stated as having been used on KCNQ channels (line 544) which is surely a typo since this study deals with Kv1.1. In addition, the statistics description is lacking. It states 1-way ANOVA was used throughout (line 552-553) but there are many figure panels that compare membrane potential for two variables, which most likely used t-tests instead.

- we have corrected the methods descriptions as requested

4) The title of the paper is misleading when it says the medicines rescue “via a novel voltage sensor binding site,” which makes it sound like a new voltage sensor site was identified in Kv1.1. Instead, from the description in the text, the voltage sensor site itself was not newly identified but rather it is a novel site for gallic acid to bind. Maybe a better more accurate phrase would be that the medicines rescue “via binding to the voltage sensor domain.”

- we have changed the title to “...via binding to the voltage sensing domain”

Reviewer #2 (Remarks to the Author):

This is a very interesting study inspired by ethnobotanical knowledge of some useful herbs in treating or helping with Episodic Ataxia 1. The authors followed rational steps first to validate the beneficial effects of some herbal extracts and then identify the major bioactive constituents mediating the desired enhancement of the EA1-relevant KCNA1 mutants. The manuscript is very well written and thoroughly enjoyable and the findings will be attractive to readers from a diverse field including ion channels, ethnopharmacology/alternative medicine, drug discovery, neuroscience etc.

I have the following points:

Major point:

Whilst it is great that the authors could identify potential binding sites via which Gallic acid was acting, it would have been great to have some thoughts on possible molecular mechanism of why binding to such region could enhance KCNA1 current. Also it is commendable that they have used an unbiased docking approach and experimentally validated key residues, nevertheless the docking was done against a static structure of the channel protein. I recommend that the authors consider:

- MD simulation to evaluate how stable the docked pose of gallic acid was.

- MD simulation of the unliganded and docked KCNA1 mutant proteins and comparative analyses of the

simulation trajectories which could potentially highlight the molecular mechanism of gallic acid's action in this regard.

- **We formed a new collaboration with the Tobias group to address these comments. We now include Molecular Dynamics simulations in Figure 10 and new Supplementary Videos 1 and 2, and describe the results and their interpretation in terms of potential mechanisms of gallic acid's effects, on page 15. We truly appreciate this suggestion as it independently supported our data and improved the resolution of the predictions, and also offered a model for how gallic acid facilitates channel activation (revised Figure 10, pages 15-17, new Supplementary Videos 1 and 2).**

Minor point:

Figure 1

Plant names should be italicized

-
- **The plant species names are now italicized in the figure legend.**

In several figures, it has been mentioned that 'n indicates number of oocytes'. It should be ideal to additionally specify the number of independent days of experiments performed. Also it has been mentioned that 'statistical comparisons by one way ANOVA' - it was not clear what post hoc test was used.

- **We have now added the number of independent days of experiments (batches of oocytes) and also information on post hoc tests.**

Reviewers' Comments:

Reviewer #1:

Remarks to the Author:

In this revised manuscript, the authors were very responsive to all the reviewer comments greatly strengthening the paper. All concerns were adequately addressed including the addition of extensive new experiments and text as requested. Overall, this is a very interesting, novel, and clinically relevant study.

Reviewer #2:

Remarks to the Author:

The authors in this revised version of the manuscript has satisfactorily addressed the points I have raised. Their additional work involving MD simulations and other changes incorporated into this revised version has made it significantly better and interesting.

I am happy to recommend acceptance of the revised version for publication in Nature Communications